# Dynamic prediction of carbon prices based on the multi-frequency combined model

Yonghui Duan[1], Yingying Fan[1], Xiang Wang[2], Kaige Liu[1] and Xiaotong Zhang[1]

[1] School of Civil Engineering, Henan University of Technology, Zhengzhou City, Henan Province, China
[2] School of Civil Engineering, Zhengzhou Aviation Industry Management College, Zhengzhou City, Henan Province, China



## ABSTRACT

As a central participant and important leader in the global climate governance system, China is facing the urgent need to predict and regulate the price of carbon emissions to promote the sound development of its carbon market. In this article, a rolling prediction model based on Least Absolute Shrinkage and Selection Operator-cheetah optimization algorithm-extreme gradient boosting (Lasso-COA-XGBoost) carbon price decomposition integration is proposed to address the defects of low prediction accuracy and insufficient model stability of a single machine learning model in the carbon price prediction problem. During the modeling process, the adaptive Lasso method is first employed to select factors from 15 primary indicators of carbon prices, identifying the most important influencing factors. Next, the COA-XGBoost model is built and the parameters of the XGBoost model are optimized using the COA algorithm. Finally, the complete ensemble empirical Mode Decomposition with adaptive noise (CEEMDAM) method is utilized to decompose the residual sequence of the COA-XGBoost model and reconstruct it into high-frequency and low-frequency components. Appropriate frequency models are applied to achieve error correction, thereby constructing the combined Lasso-COA-XGBoost-CEEMDAN model. To further enhance the predictive accuracy and practicality of the model, a rolling time window is introduced for forecasting in the Hubei and Guangzhou carbon emission trading markets, ensuring that the forecasting model can adapt to market changes in real-time. The experimental results show that, taking the carbon price prediction in Hubei as an example, the proposed hybrid model has a significant improvement in prediction accuracy compared with the comparison model (XGBoost model): the RMSE is improved by 99.9987%, the MAE is improved by 99.9039%, the MAPE is improved by 99.9960%, and the $R^2$ is improved by 0.2004%, and the advantages of this hybrid model are also verified in other experiments. The results provide an effective experimental method for future carbon price prediction.

Corresponding author
Yingying Fan, 2252151908@qq.com

# INTRODUCTION

In recent years, the world has reached an unprecedented historical juncture characterized by severe environmental degradation, with the urgent challenges posed by climate change being particularly pronounced (*Qin et al., 2022*). Currently, more than 70 countries have committed to achieving net-zero carbon emissions by 2050. The Chinese government is vigorously implementing the Paris Agreement in the meantime, aiming to attain carbon neutrality by 2060 and peak carbon emissions by 2030. To promote low-carbon pathways, China launched seven nationally unified carbon emission trading markets in 2017. In this context, building and enhancing the carbon emission trading market and putting in place a sensible carbon pricing system have emerged as key tactics and essential avenues for China to move closer to these important strategic objectives. At present, the construction of China's carbon market is still in the primary stage, and the role of the pricing mechanism has not yet been fully realized. Carbon trading prices directly reflect the effectiveness of the carbon market's design and management systems, while fluctuations in carbon prices are a significant indicator of market risk. Significant price volatility may lead to instability in the carbon market, thereby affecting the market's adjustment efficiency and the ultimate effectiveness of emission reductions (*Sun & Li, 2020*). As a result, increasing the forecasting accuracy of carbon prices has gained popularity and importance in contemporary academic study.

Currently, carbon price forecasting has become an important topic of research among scholars both domestically and internationally. This article will provide a review of the relevant literature on this subject:

Firstly, the research of carbon price prediction methods: at present, there are more carbon price prediction methods, which are summarized into three main categories: statistical models, artificial intelligence models, and combination models. Regression analysis (*Du, 2023*), the autoregressive integral moving average (ARIMA) model (*Wang et al., 2024*), the autoregressive moving average (ARMA) model (*Wen & Zhibin, 2021*), and others are common statistical model techniques. The advantage of this model is that it is easy to calculate and intuitive to understand, but they tend to ignore the nonlinear characteristics in the data and have certain assumptions about the data distribution, which are influenced by subjective factors, and their prediction accuracy is not high. In contrast, artificial intelligence models with strong adaptive and nonlinear processing capabilities perform better in capturing complex changes in carbon price series. For this reason, some scholars have introduced artificial intelligence models into carbon price prediction, and the prediction accuracy has been improved. For example, *Ling & Cao (2024)* extracted the long and short-term memory features of the carbon price series for prediction by the long short-term memory (LSTM) model, which proved the unique advantage of this method in improving computational efficiency and accuracy. However, to overcome this limitation, researchers have begun to explore combined models and optimization algorithms have been applied in several applications, as a single model may not be sufficient to fully reflect the multi-dimensional characteristics of the carbon price. For example, *Feng et al. (2023)* established the Gray Wolf Optimization Algorithm Optimized Extreme Gradient Boosting

(GWO-XGBoost) prediction model to predict three carbon emission trading prices in Guangdong, Hubei, and Fujian. *Yang et al. (2020)* used the Improved Whale Optimization Algorithm (IWOA) to optimize LSTM to predict the carbon emissions trading prices in Beijing, Fujian, and Shanghai, and the indicators representing the predictive ability were significantly improved compared with LSTM. *Duan et al. (2024)* used the Whale Optimization Algorithm (WOA) to determine the optimal parameters of extreme gradient boosting (XGBoost) to predict the price of two carbon trading markets in Hubei and Guangzhou, and the proposed hybrid model always outperforms the comparison model in terms of prediction accuracy. *Wang, Zhuang & Gao (2023)* propose an extreme learning machine model based on the improved cheetah optimization algorithm (SCO-ELM), which significantly improves the accuracy and computational efficiency of lithium-ion battery remaining service life prediction by optimizing the input weights and bias parameters, reduces the root mean square error (RMSE) by more than 40% compared with the original ELM on NASA and Oxford datasets, and outperforms the optimization model of genetic algorithms in terms of generalization performance. Although the Crayfish Optimization Algorithm (COA) has shown significant advantages in complex optimization problems in machine learning parameter optimization, engineering scheduling and other fields, its application to carbon emissions trading price prediction is still in the blank stage. However, the general combination model's ability to portray the nonlinearity, non-stationarity, and multi-scale of the carbon price series is still unable to meet the high demand for prediction accuracy, and to improve the prediction effect, it is especially important to decompose the historical carbon price data for noise reduction. Commonly used carbon price decomposition methods include empirical mode decomposition (EMD) (*Yang et al., 2020*; *Zhu, 2012*), variational mode decomposition (VMD) (*Wu et al., 2023*; *Liu et al., 2023*; *Xu & Niu, 2022*), and complete ensemble empirical mode decomposition (CEEMDAN) (*Deng et al., 2024*; *Ke et al., 2023*; *Zhao & Chen, 2021*), *etc*. After comparing several decomposition methods, *Jiang, Yu & Alam (2023)* used CEEMDAN to decompose the original carbon price sequence into multiple intrinsic modal function (IMF) components and showed that the proposed model has the best performance; *Li, Zheng & Yang (2022)* constructed an integrated carbon price prediction model based on the variational mode decomposition (VMD), and the results show that the integrated model variational mode decomposition-Generalized AutoRegressive Conditional Heteroskedasticity (VMD-GARCH) can predict the carbon price of the European Union efficiently, and the prediction is the most stable in the stage of the carbon price increase in particular.

Secondly, factors affecting carbon price prediction: in their study of EU carbon market prices, *Tsai & Kuo (2014)* note the impact of coal, oil, and gas prices on carbon prices. Similarly, *Keppler & Mansanet-Bataller (2010)* used the Granger causality test combined with OLS regression analysis to demonstrate that coal and natural gas have a significant impact on EUA futures prices. *Luo et al. (2022)* focused on the Beijing carbon market and revealed a negative correlation between the carbon price and the natural gas price by using wavelet analysis. Price has a negative correlation and shows an unstable dependence on oil prices. This suggests that within the same market, the prices of different energy

commodities may have different impacts on the carbon price. On the other hand, *Zhang & Xu (2020)* used a GARCH model to illustrate the volatility characteristics of carbon prices under different economic backgrounds when analyzing data from the Shenzhen Carbon Exchange and confirmed that climate uncertainty is an important factor affecting carbon prices; *Wang et al. (2018)* showed that there is a significant correlation between air quality index (AQI) and carbon trading price in China's carbon market, which suggests that the potential impact of air quality conditions on carbon price deserves further attention. Meanwhile, *Han et al. (2019)* extended the scope of research in this area by introducing environmental factors other than energy, economic, and weather conditions into the indicator system. Together, these studies show that the factors affecting the carbon price are multifaceted, including, but not limited to, energy price changes, macroeconomic conditions, climate change trends, and local environmental quality levels.

In summary, although the existing carbon price forecasting studies have made significant progress, they still have the following limitations. First, after decomposing the forecasting errors, the existing studies usually use a single forecasting model to forecast each subsequence, failing to fully consider the heterogeneity of different subsequences. For example, *Li, Zheng & Yang (2022)*, after decomposing the historical price of carbon price by VMD, use the same model to forecast all residual series, ignoring the unique fluctuation patterns of different frequency band subsequences; second, when dividing the dataset, the existing studies usually directly divide the original time series into training and testing sets proportionally, which implicitly assumes that the future data are available, which may lead to data leakage and affect the the practical application effect of the model (*Qian et al., 2019*; *Du, Zhao & Lei, 2017*); third, existing studies are less likely to incorporate public attention (*e.g.*, Baidu index) into the prediction model. Although some studies have confirmed the influence of exogenous factors on carbon prices (*Li & Lei, 2018*; *Zhou & Li, 2019*), these studies mainly focus on the direct relationship between energy prices, macroeconomic and environmental factors and carbon prices, and ignore the market expectations and risk preferences that may be reflected by public attention.

To overcome the above limitations, this article proposes a new hybrid model for carbon price prediction, the Lasso-COA-XGBoost-CEEMDAN model, based on previous research. The model incorporates the adaptive Lasso method, the cheetah optimization algorithm (COA), extreme gradient boosting (XGBoost), and complete ensemble empirical mode decomposition with adaptive noise (CEEMDAN) techniques. First, the adaptive Lasso method is used to screen the key factors affecting the carbon price; second, the COA-XGBoost model is constructed and the parameters of the XGBoost model are optimized using the COA algorithm; then, the residual sequences of the COA-XGBoost model are decomposed using the CEEMDAN method, and the differentiated prediction model is constructed for the subsequences with different frequencies to perform error correction; finally, the predicted values are added with the corrected residuals to obtain the final carbon price prediction results.

The main contributions of this article are as follows: (1) a differentiated forecasting method for different frequency residual series is proposed, which improves the accuracy of error correction; (2) an integrated forecasting framework for dynamic carbon price

decomposition based on rolling time windows is constructed, which avoids the data leakage problem that may be caused by the traditional static model; and (3) the Baidu index is incorporated into the forecasting model, which provides a more comprehensive source of information for the prediction of the carbon price and helps to capture the impact of market sentiment and expectations on carbon price.

# ALGORITHM INTRODUCTION

## Feature selection

Feature selection becomes a critical tool when dealing with large, high-dimensional datasets. It helps identify the key features that most significantly influence prediction outcomes, effectively reducing redundancy in the model and enhancing predictive accuracy (*Hao & Tian, 2020*). Additionally, feature selection contributes to improving the interpretability of the model, providing a more solid foundation and support for data analysis and decision-making. This study employs the partial autocorrelation function (PACF) to determine the optimal historical carbon price data and utilizes the Lasso regression (LASSO) method to select the most relevant external influencing factors.

### PACF

Partial Autocorrelation Function (PACF) is a tool used in time series analysis to measure the partial correlation between a time series and its own lagged values, specifically the correlation between the current time point and a specified lagged time point after removing the influence of other lagged factors. More specifically, represents the conditional correlation between and after removing the effects of the intervention variable, which is the partial autocorrelation between and *Hao & Tian (2020)*.

The values of PACF are typically calculated using the Yule-Walker equations, and these computed values are used to plot the PACF graph. In the PACF graph, lag orders that exceed the confidence interval may have a significant impact on the target variable, while the partial autocorrelation coefficients for other lag orders approach zero, suggesting that these lag orders may not significantly influence the prediction of the target variable (*Li et al., 2021*). Additionally, to correctly interpret the results of the PACF graph, it is essential to ensure the stationarity of the time series.

### Lasso

Lasso regression, first proposed by British Robert Tibshirani, is to prevent overfitting and solve the problem of severe covariance by generating a penalty function that is a compression of variable coefficients in the regression model, which is currently used very widely in forecasting and prediction models. modeling is very widely used, and its principle is as follows (*Li et al., 2023*).

Let $x$ be the independent variable and $y$ the dependent variable, with standardized values of the observed data obtained from $m$ samples represented as $(x, y)$, where $x$ is an $m \times k$ matrix (with $m > k$), and $y$ is an $m \times 1$ matrix. The i-th observation of $x$ is given by $x_i = (x_{i1}, x_{i2}, \ldots, x_{ik})^T$, where $i \in [1, 2, \ldots, m]$ and all observations are independent. The regression model of on is expressed as Eq. (1).

$$y_i = \widehat{\alpha} + \sum \beta_j x_{ij} + \varepsilon_i. \tag{1}$$

In the equation $\varepsilon_i - N(0, \sigma^2)$; $\hat{a} - = \bar{y}$, the standardized data indicates that, $\bar{y} = 0$. After rearranging Eqs. (1), (2) is obtained.

$$y = \beta x + \varepsilon. \tag{2}$$

In the equation, $\varepsilon_i - N(0, \sigma^2)$; $\beta - n$ represents the parameter vector, and $\varepsilon-$ the random disturbance term. To select the significant influencing variables, we need to add a constraint to Eq. (2), as expressed in Eq. (3).

$$\arg \min_{\{\beta_1, \beta_2, \cdots, \beta_m\}} \| y - \beta x \|^2 \qquad \text{s.t} \sum_j \frac{|\beta|}{\sum \beta_j^0} \leq s. \tag{3}$$

In the equation, $s-$is equal to a value within the range $[0, 1]$; the $t-1$ harmonic parameters are $\geq 0$.

Lasso regression works by continuously adjusting the $t$ value to reduce the overall regression coefficients of the model, systematically shrinking the coefficients of insignificant variables until they reach 0.

## Cheetah optimization algorithm

COA is a new type of group intelligence optimization algorithm, which is proposed by *Amin et al. (2022)* in 2022 as a new type of group intelligence optimization algorithm inspired by cheetah hunting in nature, which achieves the position updating by simulating the three strategies of cheetahs' searching, sitting and waiting, and attacking in the process of hunting, and it has the strong ability of searching for the optimal, and the quick speed of convergence and other characteristics.

Research indicates that cheetahs typically succeed in capturing their prey within 30 s, earning them the title of the "fastest land animal." In 2022, *Amin et al. (2022)* introduced a nature-inspired meta-heuristic algorithm for large-scale optimization problems known as the COA. This algorithm simulates the hunting mechanism of cheetahs, with its mathematical model encompassing three strategies: search, wait, and attack.

### *Search strategy*

$$x_{i,j}^{t+1} = x_{i,j}^t + r_{i,j}^{-1} \alpha_{i,j}^t \tag{4}$$

In the expression, t represents the current hunting time; $x_{i,j}^{t+1}$ and $x_{i,j}^t$ denote the updated position and current position of cheetah i (where $i = 1, 2, \ldots, N$) in arrangement j (where $j = 1, 2, \ldots, D$), with N being the number of cheetahs in the population and $D$ representing the dimensionality of the optimization problem. $r_{i,j}$ and $\alpha_{i,j}^t$ are the randomization parameter and step size for the cheetah i in the arrangement j, respectively, where $\alpha_{i,j}^t > 0$, is typically set to $0.001 \times \frac{t}{T}$, and T is the maximum duration of the hunting time.

### Waiting strategy

In this mode, to avoid alerting the prey to their presence, cheetahs maintain their position and wait for the prey to approach. This behavior can be modeled as follows:

$$x_{i,j}^{t+1} = x_{i,j}^{t}. \tag{5}$$

In the equation, the meanings of the parameters remain the same as previously described. This strategy helps prevent the COA from converging too early.

### Attack strategy

At this stage, cheetahs utilize their speed and agility to capture prey. In group hunting, each cheetah can adjust its position based on the fleeing prey and the status of the leader or nearby cheetahs. These attack strategies are mathematically defined as follows:

$$x_{i,j}^{t+1} = x_{B,j}^{t} + \theta_{i,j} \beta_{i,j}^{t}. \tag{6}$$

In the equation, $B$ represents the prey; $x_{B,j}^{t}$ denotes the current position of the prey in arrangement $j$; $\theta_{i,j}$ and are the $\beta_{i,j}^{t}$ turning factor and interaction factor related to cheetah i in arrangement $j$, respectively. Here, $\theta_i$ is a random number equal to $|b_{i,j}|^{exp(b_{i,j}/b_{i,j})} \sin(2\pi b_{i,j})$, and $b_{i,j}$ is a random number following a standard normal distribution.

In the study by *Amin et al. (2022)* 14 CEC-2005 benchmark functions were thoroughly tested. The set of benchmark functions covers a wide range of optimization problems with different characteristics, which can effectively test the performance of the algorithm in different scenarios. The results show that the COA algorithm outperforms the classical state-of-the-art algorithms such as Differential Evolutionary algorithm (DE), Gray Wolf Optimization algorithm (GWO), genetic algorithm (GA), and particle swarm optimization algorithm (PSO) in nine of them in terms of both the mean and the standard deviation. Further, when dealing with multiple classical optimization problems as well as large-scale optimization problems, the COA algorithm also outperforms other competing algorithms in all key performance metrics. This demonstrates the powerful optimization capability, efficient convergence speed and good stability of COA algorithm in complex optimization tasks. In view of this, it shows great potential in optimizing complex models. The XGBoost model in this study, as a widely used and high-performance machine learning model, still has room for further optimization when facing specific complex tasks. Therefore, considering the advantages of the COA algorithm and the optimization requirements of the XGBoost model, this article decides to choose the COA algorithm to optimize XGBoost.

## COA-XGBoost model

XGBoost is a boosting-type model developed by *Chen & Guestrin (2016)* in 2016, which combines linear solvers with learning algorithms for classification and regression trees. The fundamental idea of this model is to combine multiple decision tree models with lower

predictive accuracy using a specific strategy to create a more accurate ensemble model. During the model training process, XGBoost iteratively optimizes the model through gradient boosting, generating a new decision tree in each iteration to fit the residuals produced in the previous iteration. Through this iterative optimization approach, XGBoost can continuously enhance the model's predictive accuracy and generalization capability. The traditional gradient boosting decision tree (GBDT) method only utilizes the first derivative, while XGBoost performs a second-order Taylor expansion of the loss function, incorporating a regularization term to control model complexity and mitigate overfitting. Additionally, XGBoost employs a more refined evaluation method at the split nodes, enabling it to better capture the nonlinear relationships between features. In recent years, the XGBoost model has demonstrated strong performance in various fields, including financial risk control, healthcare, and natural language processing. The mathematical principles of this model are as follows:

Define the ensemble model of trees as follows:

$$\hat{y}_i = \sum_{m=1}^{M} f_m(x_i), f_m \in F. \tag{7}$$

In the equation, $\hat{y}_i$ represents the predicted value; $M$ is the number of decision trees; $F$ denotes the space of tree selections; and $x_i$ is the i-th input feature.

The loss function of the XGBoost model is given by:

$$L = \sum_{i=1}^{n} l(y_i, \hat{y}_i) + \sum_{m=1}^{M} \theta(f_m). \tag{8}$$

In this equation, the first part of the function represents the prediction error between the XGBoost model's predicted values and the actual training values, while the second part reflects the complexity of the trees. The main purpose of this second part is to serve as regularization to control the model's complexity:

$$\theta(f_m) = \gamma T + \frac{1}{2}\tau \|\omega\|^2. \tag{9}$$

In the equation, $\gamma$ and $\tau$ are penalty factors. During the process of minimizing the loss function, incorporating the incremental function $f_t(x_i)$ in Eq. (9) can minimize the value of the loss function to the greatest extent. Thus, the objective function for the t-th iteration is given by:

$$L^{(t)} = \sum_{i=1}^{n} l(y_i, \hat{y}_i) + \sum_{m=1}^{M} \theta(f_m) = \sum_{i=1}^{n} l(y_i, \hat{y}_i^{t-1} + f_t(x_i)) + \theta(f_t). \tag{10}$$

At this point, a second-order Taylor expansion of Eq. (11) is performed to approximate the objective function, and the sample set in each leaf of the j-th tree is defined as $I_j = \{i|q(x_i = j)\}$. The approximation can be represented as follows:

$$L^{(t)} \cong \sum_{i=1}^{n}\left[g_i f_t(x_i) + \frac{1}{2}h_i f_t^2(x_i)\right] + \theta(f_t) \cong \sum_{i=1}^{n}\left[g_i f_t(x_i) + 1/2 h_i f_t^2(x_i)\right] + \gamma T + \frac{1}{2}\tau\omega^2$$

$$\cong \sum_{j=1}^{T}\left[\left(\sum_{i\in I_j}g_i\right)\omega_j + \frac{1}{2}\left(\sum_{i\in I_j}h_i + \tau\right)\omega_j^2\right] + \gamma T. \tag{11}$$

In this equation, $g_i = \partial_{\hat{y}_i^{t-1}}l(y_i, \hat{y}_i^{t-1})$ represents the first derivative of the loss function, and $h_i = \partial_{\hat{y}_i^{t-1}}^2 l(y_i, \hat{y}_i^{t-1})$ denotes the second derivative of the loss function. Defining $G_i = \sum_{i\in I_j}g_i$ and $H_i = \sum_{i\in I_j}h_i$, the following is obtained:

$$L^t \cong \sum_{j=1}^{T}\left[G_j\omega_j + \frac{1}{2}\left(H_j + \tau\right)\omega_j^2\right] + \gamma T. \tag{12}$$

Taking the partial derivative with respect to $\omega$, the following is obtained:

$$\omega_j = -\frac{G_j}{H_j + \tau}. \tag{13}$$

Substituting the weights into the objective function, the following is obtained:

$$L^{(t)} \cong -\frac{1}{2}\sum_{j=1}^{T}\frac{G_j^2}{H_j + \tau} + \gamma T \tag{14}$$

In the training process of the XGBoost model, the choice of different parameters can significantly impact the prediction results; thus, the model's performance is largely determined by the selection of parameters. There are a total of 23 hyperparameters in the XGBoost algorithm, which are primarily divided into general parameters for controlling the overall function, booster parameters for managing the details of the booster and learning objective parameters that control the training objectives. The COA-XGBoost combined model uses the five hyperparameters that most significantly affect performance in XGBoost (namely, learning_rate, subsample, colsample_bytree, max_depth, and alpha) as the position vector $\alpha$ of the cheetah in the COA algorithm. Through iterative updates *via* the COA algorithm, these parameters are continuously optimized until the global optimal position is output as the final parameters for the XGBoost model.

## CEEMDAN model

The CEEMDAN algorithm, proposed by *Torres et al. (2011)* in 2011, is a data-driven algorithm used for signal processing and analyzing nonlinear and adaptive signals. The basic idea is that the original signal is decomposed into a series of IMF and the algorithm parameters are adjusted using adaptive noise, which improves the quality and stability of

the decomposition. Compared to the traditional EMD algorithm, the decomposition steps of CEEMDAN are more stable and can reduce randomness in the decomposition process by integrating the results of multiple decompositions. In contrast to the ensemble empirical mode decomposition (EEMD) algorithm, CEEMDAN adds adaptive white noise at each stage, overcoming the significant reconstruction error associated with the EEMD method. CEEMDAN performs exceptionally well in handling complex signals and reducing data fluctuations, making it suitable for addressing the issue of carbon trading price prediction.

The steps of the CEEMDAN decomposition can be summarized as follows:

**Step 1:** Add an adaptive Gaussian white noise sequence $\sigma n_i(t)$ to the original sequence $y(t)$ to obtain a new sequence $\bar{y}_i(t)$ with noise:

$$\bar{y}_i(t) = y(t) + \sigma n_i(t), i = 1, 2, \cdots N \tag{15}$$

where $n_i(t)$ represents the added white noise, and the intensity of the noise sequence is determined by parameter $\sigma$.

**Step 2:** Perform a complete EMD on the noisy original signal to obtain a set of IMFs and a residual component:

$$\mathrm{imf}_1(t) = \frac{1}{N} \sum_{i=1}^{N} \mathrm{imf}_{1i}(t) . \tag{16}$$

At this point, the calculation formula for the residual component $R_1(t)$ is:

$$R_1(t) = \bar{y}_i(t) - \mathrm{imf}_1^{'}(t) . \tag{17}$$

**Step 3:** The residual is taken as the new input data, and the adaptive white noise sequence $\sigma n_i(t)$ is added to $R_1(t)$ to obtain new input data $R_1(t) + \sigma E_1(n_i(t))$. *Here, $E_j(\cdot)$ is the $j$-th intrinsic mode function obtained after EMD decomposition.* At this point, the new sequence undergoes EMD decomposition and averaging to obtain the second mode component and the residual component:

$$\mathrm{imf}_2(t) = \frac{1}{N} \sum_{i=1}^{N} E_1(R_1(t) + \sigma_1 E_1(n_i(t))) \tag{18}$$

$$R_2(t) = R_1(t) - \mathrm{imf}_2^{'}(t) \tag{19}$$

**Step 4:** Repeat Steps 1, 2, and 3 to ultimately obtain the $j+1$-th mode component and the $j$-th residual component:

$$\mathrm{imf}_{j+1}(t) = \frac{1}{N} \sum_{i=1}^{N} E_1(R_j(t) + \sigma_j E_j(n_i(t))) \tag{20}$$

$$R_j(t) = R_{j-1}(t) - \mathrm{imf}_j^{'}(t) \tag{21}$$

**Step 5:** The above steps are repeated until the residual component can no longer be subjected to the EMD decomposition. Finally, the original sequence will be resolved into several intrinsic mode functions and a trend component.

$$y(t) = \sum imf(t) + R_{es}(t). \qquad (22)$$

After the CEEMDAN has completed the decomposition of the original sequence, appropriate predictive models are applied to each intrinsic mode function component for individual predictions, and the results of each component are aggregated to obtain the final residual prediction results.

## COA-XGBoost-CEEMDAN model

This study aims to improve the prediction accuracy of carbon trading prices by proposing a hybrid algorithm based on CEEMDAN, COA, and XGBoost. The main steps of the experiment are as follows:

First, the COA algorithm is used to optimize the parameters in the XGBoost model, and the optimized model is used for the initial forecast. Next, CEEMDAN is used to decompose the residual carbon price series, which helps to understand the characteristics and variation patterns of the signal. The decomposed IMFs are then split into high-frequency and low-frequency components, with the COA-XGBoost model and the SVR model used to predict the different frequency components, respectively. Finally, the predicted values of the different mode components are summed with the initial predicted values to obtain the final prediction results of the model. The flowchart is shown in Fig. 1.

In the entire prediction process, COA-XGBoost is used for predictions at two different stages. The considerations for its application at each stage are as follows: first, in the first stage, complex and high-dimensional datasets need to be processed, which puts higher requirements on the robustness and computational efficiency of the model. Based on this, the COA-optimized XGBoost algorithm is selected as the prediction model for the first stage due to its strong fitting ability to high-dimensional nonlinear data. Second, the prediction task in the second stage is relatively straightforward, as the residual sequence obtained from the decomposition of the original data has effectively removed noise and irregular fluctuations, providing a more solid and precise data foundation for subsequent predictions. At this stage, these residuals can be further differentiated into high-and low-frequency components, and the most suitable prediction model can be selected for different frequency data. Since the high-frequency residuals still contain rich and valuable information, and to maintain the overall coherence of the research, this article uses the COA-XGBoost method to handle the prediction of the high-frequency part.

# EXPERIMENTAL DATA AND EVALUATION STANDARDS
## Establishment of primary indicator system
### Carbon market and its trading prices
The carbon emissions trading market is a market that combines total control with market trading, characterized by a clear policy orientation. It aims to promote emissions reduction

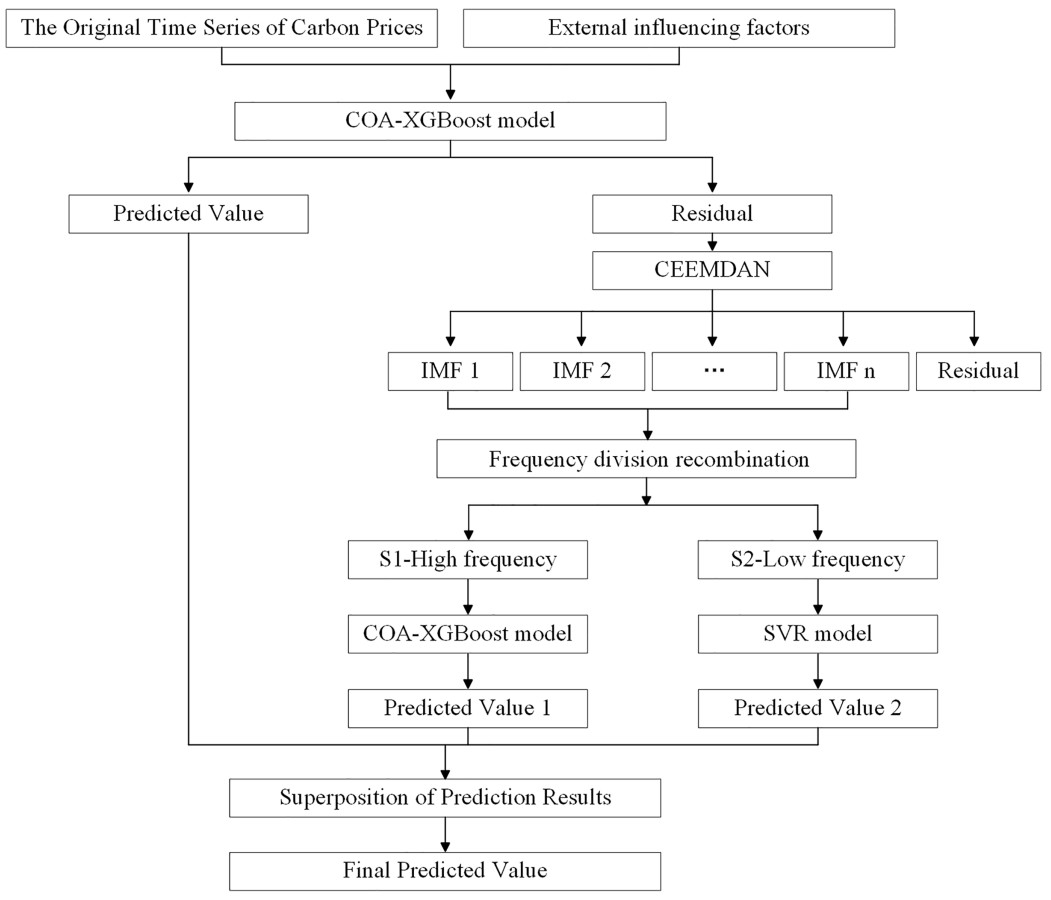

**Figure 1 The framework of the carbon price forecasting based on COA-XGBoost-CEEMDAN.**

through economic incentive mechanisms, helping to achieve greenhouse gas reduction targets. Governments or regulatory agencies set an overall emissions cap and allocate this cap to various emitting units. These units can meet their emissions requirements through voluntary reductions, purchasing carbon credits, or engaging in carbon offsets. After years of practice and development, the total transaction volume of the eight pilot markets in China has generally shown a steady growth trend, except for a decline in 2020 due to the pandemic. Under the guidance of the "dual carbon" goals, these markets have gradually formed trading mechanisms that align with their development characteristics. In terms of quota setting and allocation, a gradually decreasing control coefficient has been introduced, and efforts are being made to explore a compensated quota trading mechanism to enhance the carbon market's price discovery function.

In Table 1 it can be seen that Hubei and Guangzhou carbon emission rights exchanges have significant advantages in terms of trading volume, trading turnover, market mechanism, and innovative initiatives, reflecting their important position and influence in China's carbon market (*Liu, Jiang & Ye, 2020*; *Liu, Ma & Wang, 2015*). Therefore, this article collects Hubei and Guangzhou carbon market data from CHOICE Financial

**Table 1 Comparative data of eight major carbon emission rights exchanges in China.**

| Carbon exchange | Area | Established year | Annual trading volume (10,000 tons) | Annual trading amount (100 million yuan) | Main traded varieties | Trading mechanism | Policy support | Innovative measures |
|---|---|---|---|---|---|---|---|---|
| Beijing | North China | 2013 | 500 | 50 | Carbon emissions rights | Spot trading | Strong | Few |
| Shanghai | East China | 2013 | 600 | 60 | Carbon emissions rights | Spot trading | Strong | Few |
| Tianjin | North China | 2013 | 550 | 55 | Carbon emissions rights | Spot trading | Strong | Few |
| Chongqing | South-west | 2013 | 450 | 45 | Carbon emissions rights | Spot trading | Medium | Few |
| Shenzhen | South China | 2013 | 700 | 70 | Carbon emissions rights | Spot trading | Strong | Few |
| Fujian | East China | 2016 | 300 | 30 | Carbon emissions rights | Spot trading | Medium | Few |
| **Guangzhou** | **South China** | **2013** | **800** | **80** | **Carbon emissions rights, carbon forwards** | **Spot, forward** | **Strong** | **Many** |
| **Hubei** | **Central China** | **2014** | **750** | **75** | **Carbon emissions rights** | **Spot, auction** | **Strong** | **Many** |

**Note:**
Data source: annual reports of the exchanges (2023). Results with significant advantages are shown in bold.

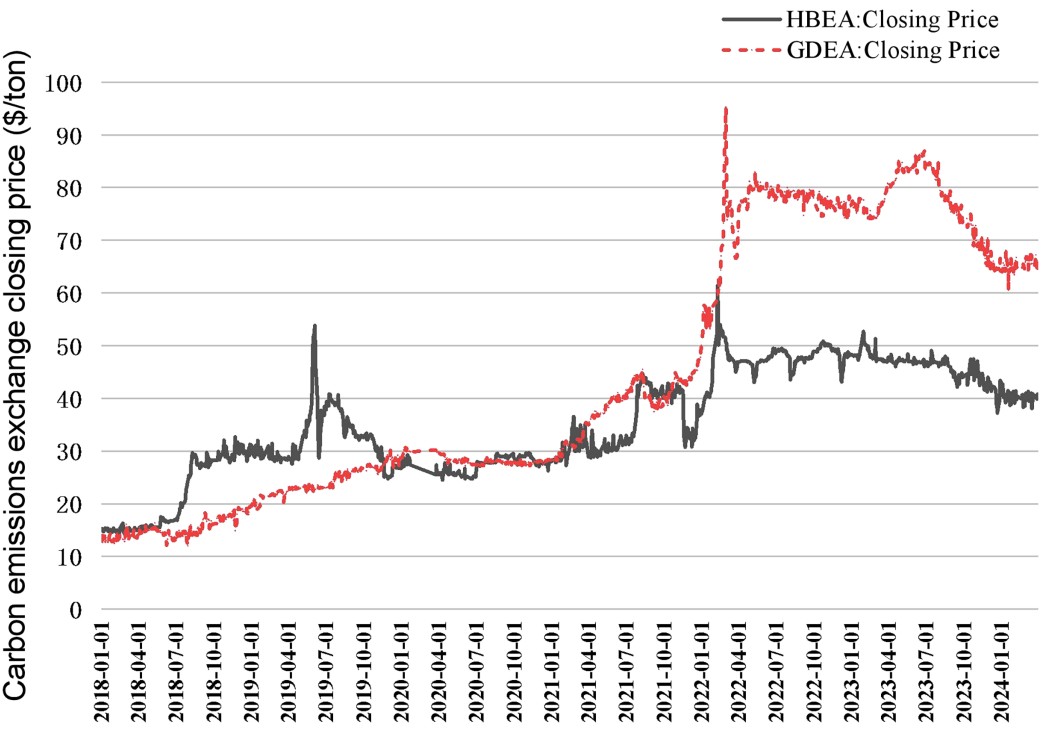

**Figure 2 Closing price of carbon emission rights in Hubei and Guangzhou carbon markets from 2018 to 2024.**

Terminal (Fig. 2), which helps to comprehensively understand and analyze the carbon emission right trading price and its influencing factors.

### Carbon market and its trading prices

The volatility of carbon prices is complex and is influenced by both internal and external factors. The internal factors are the historical trend of the carbon price, and the external influences are the external factors that are closely related to the carbon price. Concerning previous literature and following the principles of comprehensiveness, objectivity, and data availability, this article selects a total of 16 indicators to build a primary indicator model of carbon price influencing factors (as shown in Table 2) from six dimensions, namely, international carbon price, macroeconomics, energy price, climatic conditions, macro-policy, and emerging media indexes, with the specific choices and bases as follows:

International carbon price: The EU carbon market is recognized as one of the most mature carbon markets in the world, while China's carbon market, despite its rapid development in recent years, still has a certain gap with the EU carbon market in terms of market maturity (Yi et al., 2022). Therefore, in terms of institutional design, China usually draws on the experience of the EU carbon market (Li & Song, 2022). In this study, the EUA futures settlement price (Huang & He, 2020; Wang, Cheng & Sun, 2022) (denoted as V1) of the EU carbon market is chosen as a representative indicator of the international carbon price, to reflect the price dynamics of the international carbon market.

Macroeconomics: There is an interaction mechanism between the output of different industries and the overall economic situation, while the arbitrage between the domestic and international carbon markets also causes fluctuations in the price of carbon emissions (Sun, Hao & Li, 2022). As a large industrialized country, China is in the process of transitioning from traditional industry to green and sustainable development. As a result, its demand for carbon emissions is still high (Qin et al., 2018). Therefore, in this article, the SSE Composite Index (denoted as V2) is selected to represent China's industrial development, and the CSI 300 Index (denoted as V3) is selected to represent China's macroeconomic situation. In addition, the stock index of Germany, the largest economy in the European Union, i.e., Frankfurt DAX (denoted as V4) is selected to represent the economic situation of the European Union, and the Standard & Poor's 500 Index (denoted as V5) is selected to represent the economic situation of the United States.

Energy prices: Coal, crude oil, and natural gas are known as the world's three major energy sources, and changes in energy prices will have a direct impact on the production costs of the relevant enterprises, but also may prompt enterprises to shift to lower-cost energy options or adopt more efficient emission reduction technologies, which in turn affects the demand for carbon emissions from these enterprises and ultimately affects the changes in carbon prices (Xie et al., 2022). In recent years, several studies have demonstrated the influence of energy prices on carbon trading prices (Xu et al., 2022; Alberola, Chevallier & Chèze, 2008; Chevallier, Nguyen & Reboredo, 2019; Christiansen et al., 2011; Ji, Zhang & Geng 2018; Mansanet-Bataller, Pardo & Valor, 2007). In this article, the coking coal (V6) and power coal (V7) futures settlement prices are selected to represent the coal price in China, the Brent crude oil futures settlement price (V8) represents the

**Table 2  Primary index model of carbon price influencing factors.**

| Indicator layer | | Serial number | Sources of data |
|---|---|---|---|
| **Primary indicators** | **Secondary indicators** | | |
| International carbon price | EU Emission Allowances (EUA) | V1 | Choice |
| Macro-economic | SSE (Shanghai Stock Exchange) Composite Index | V2 | Choice |
| | CSI 300 index (stock market index) | V3 | Choice |
| | DAX Frankfurt, Germany | V4 | Choice |
| | Standard and Poor's 500 index (S&P 500) | V5 | Choice |
| Energy price | Coking Coal Futures Settlement Price | V6 | Choice |
| | Power Coal Futures Settlement Price | V7 | Choice |
| | Brent crude oil futures settlement price | V8 | Choice |
| | NYMEX Natural Gas Futures Close | V9 | Choice |
| Climatic conditions | Air Quality Index (AQI) | V10 | Choice |
| Macroeconomic policy | USD/CNY Mid Price (USDCNY) | V11 | Choice |
| | EUR/CNY Mid Price (EURCNY) | V12 | Choice |
| Emerging Media Index | Baidu Search Index-Carbon Neutral | V13 | Choice |
| | Baidu Search Index-Carbon Trading | V14 | Choice |
| | Baidu Search Index-Carbon Peak | V15 | Choice |
| | Baidu Search Index-Carbon Sink | V16 | Choice |

international crude oil market price, and the NYMEX natural gas futures closing price (V9) represents the international natural gas market price.

Natural environment: The occurrence of extreme weather events usually leads to an increase in energy consumption, such as the need for more cooling or heating, which can increase carbon emissions and thus push up the price of carbon credits (*Alberola, Chevallier & Chèze, 2009*; *Considine, 2000*). In addition, deteriorating air quality may prompt the government to strengthen the regulation of carbon emissions (*Ji et al., 2021*; *Zheng, Song & Shen, 2021*). Therefore, in this article, the air quality index (AQI) (*Creti, Jouvet & Mignon, 2012*; *Han et al., 2019*) (V10) of Wuhan and Guangzhou, the locations of the carbon markets in Hubei and Guangzhou, respectively, are selected as variables of the natural environment to be included in the model.

Macro policy: Exchange rate fluctuations will have a direct impact on international trade, which in turn will affect companies' production activities. Therefore, this article selects the dollar (V11) and the euro (V12) against the yuan median price represents the exchange rate.

Emerging media indicators: Considering that Baidu is a website that people often use to get information, the Baidu index is also selected as an influential factor for research. After searching and screening, four keywords, namely "carbon neutral", "carbon trading", "carbon peak" and "carbon sink", were finally identified to represent the emerging media index. The keywords "carbon neutral", "carbon trading", "carbon peak" and "carbon sink" represent the emerging media index.

## Variable selection and data sources

In this article, the carbon market price of Hubei Province and Guangzhou City is selected as the dependent variable, and the historical carbon price as well as a variety of domestic and international influencing factors are used as independent variables for empirical analysis. The data used covers the daily data from January 1, 2018, to March 31, 2024, which are all derived from the Choice Financial Terminal database. In the process of data selection, the effects of domestic and international public holidays, differences in trading hours, and missing values of variables are especially considered. In addition, to extract the time series characteristics of the carbon price data, this article converts the raw data into a stacked data type and incorporates the sliding window method for forecasting. Specifically, assuming that the target of prediction is the carbon price at the t+3th moment, the input data will include continuous observations from the t-10th moment to the t-th moment (*Li, Liang & Zhou, 2016*). In this study, the size of the sliding window is set to five quarters, *i.e.*, data from the first five quarters (January 1, 2018, to March 31, 2019) are used as the training set to predict the carbon price in the sixth quarter (April 1, 2019, to June 30, 2019). A series of overlapping sample datasets are created by moving the data backward quarter by quarter, with each move having a step size of one quarter. The exact structure of the sliding window is shown in Fig. 3.

## Data pre-processing

There are different magnitudes and units between the collected data on the impact factors, to make the data more comparable, the data need to be linearly normalized to convert the data to the same magnitude or unit, so that it can be easily compared and analyzed and also can improve the accuracy and effectiveness of the machine learning algorithms. The expression for normalization is as follows:

$$x^* = \frac{x - x_{min}}{x_{max} - x_{min}}. \tag{23}$$

Among them, $x^*$ represents the normalized value; x is the original data, $x_{min}$ is the minimum value in the dataset, and $x_{max}$ is the maximum value in the dataset. After normalization, the data falls within the range of [0, 1].

## Establishment of the ultimate indicator system

The economists David Dickey and Wayne Fuller proposed the Augmented Dickey-Fuller (ADF) test in 1979. The test is a statistical method used to test whether the time series data has a unit root, *i.e.* to verify whether the data is smooth. Through the ADF test, it can be obtained that the *p*-value of Hubei is 0.252318 and the *p*-value of Guangzhou is 0.766068, both of which are larger than the usually chosen significance level (*e.g.*, 0.05 or 0.01), and therefore the original hypothesis cannot be rejected, *i.e.*, neither of the two carbon markets' historical carbon price data has smoothness. After one difference was performed separately, the data had all been smooth, and partial autocorrelation analysis was subsequently introduced to select the input historical characteristics for the prediction method. Figure 4 demonstrates the PACF results, and it can be seen that both Hubei and

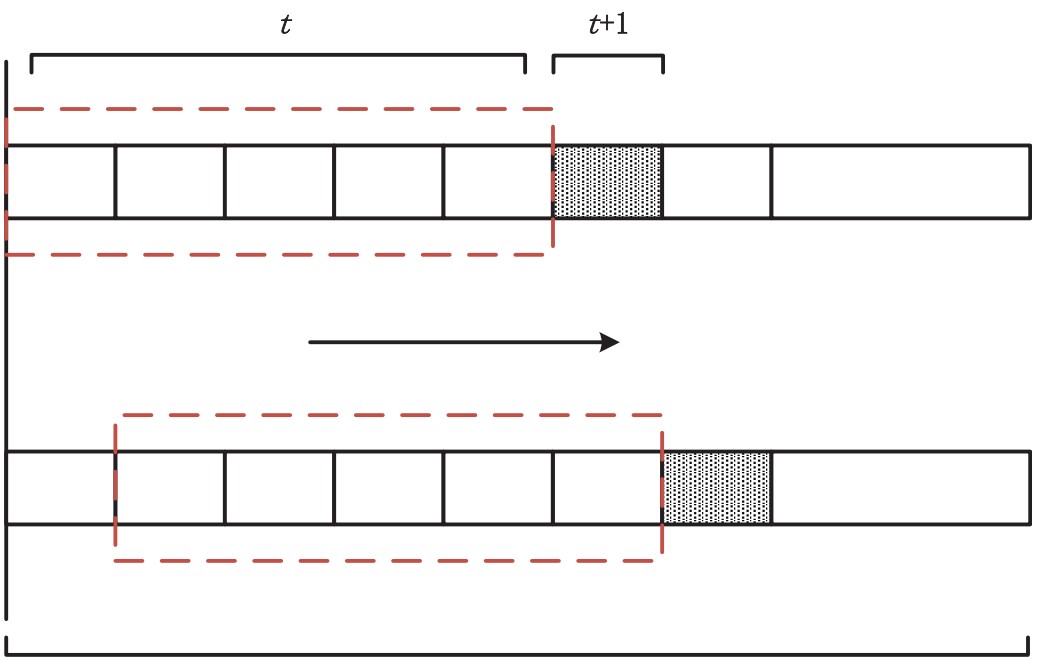

Full data

**Figure 3  Sliding window structure diagram.**       

Guangzhou carbon price data have significant third-order autocorrelation. $x_i$ is the output feature, and $\{x_{i-1}, x_{i-6}, x_{i-8}\}$ is determined as the input historical variable of Hubei carbon price. $\{x_{i-1}, x_{i-4}, x_{i-5}\}$ is identified as the input historical variable of Guangzhou carbon price.

More input features may reduce the prediction accuracy due to redundancy. Identifying and screening out the most effective variables can not only achieve the goal of dimensionality reduction but also effectively avoid the problem of multicollinearity among data, thus improving the prediction performance of the model. Therefore, to identify the main external influences affecting carbon prices in Hubei and Guangzhou, this article adopts the Lasso regression algorithm for feature selection.

After normalizing all the data, Lasso regression analysis was carried out with the 16 external influences listed in Table 2 as independent variables and carbon emissions as dependent variables, with the parameter K value taken as 0.01. The correlation analysis shows that the top 10 variables in terms of carbon price correlation in Hubei Province are the S&P 500, Baidu index-Carbon Neutral, Coking Coal Futures Clearing Price, Power Coal Futures settlement price, EURCNY, Brent crude oil futures settlement price, NYMEX natural gas futures closing price, AQI, Baidu index-carbon trading, CSI 300 index, and the top 10 carbon price-related variables in Guangzhou are EUA, Brent crude oil futures settlement price, USDCNY, Power Coal Futures Settlement Price, Baidu index-Carbon Peak, S&P 500 Index, SSE Composite Index, EURCNY, Coking Coal Futures Settlement Price, and Baidu search index-Carbon Sinks, as shown in Fig. 5, whereby they are identified

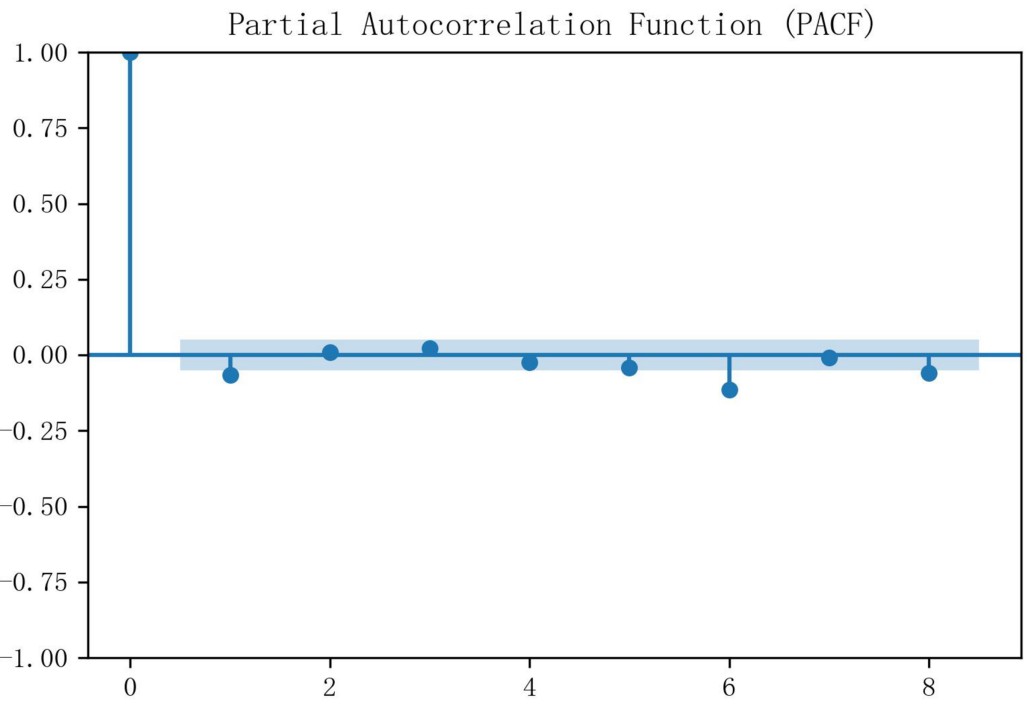

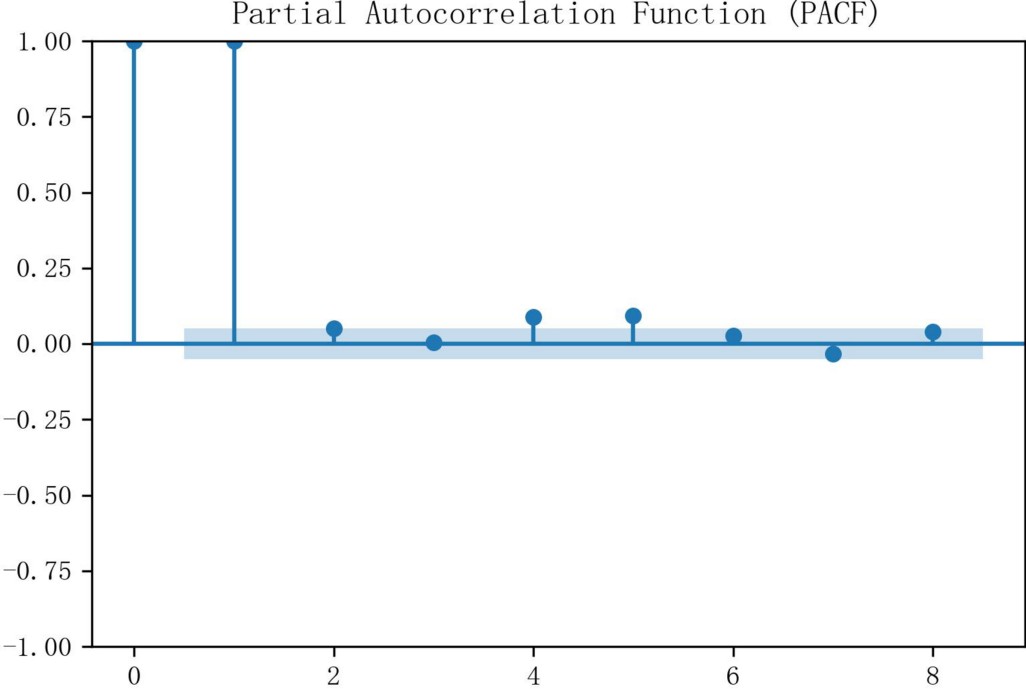

**Figure 4  PACF results of carbon price data in Hubei and Guangzhou.**

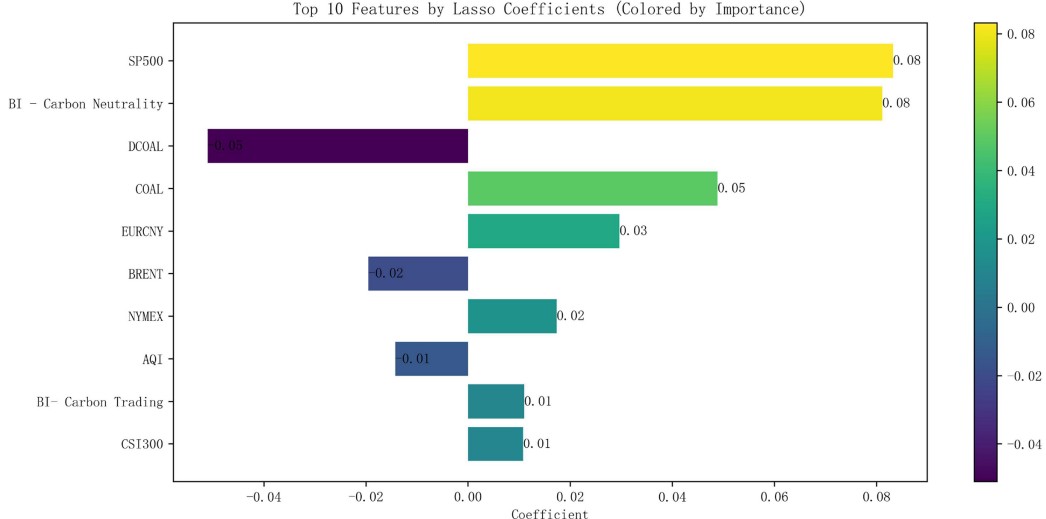

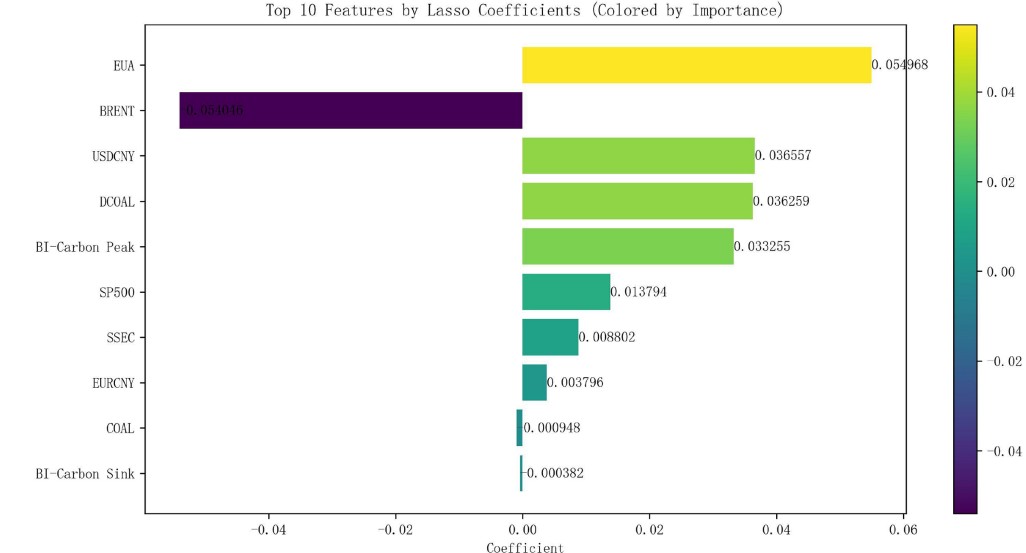

**Figure 5 Lasso results of external factors in Hubei and Guangzhou.**

as the 10 key influencing factors affecting the carbon price in Hubei Province and Guangzhou City, respectively.

## Evaluation indicators

In this article, mean square error (MSE), mean absolute error (MAE), mean absolute percentage error (MAPE) and coefficient of determination (R-squared or $R^2$) are used as indicators to assess the prediction performance of the model. The MSE is the average of the square of the differences between the predicted values and the actual values, while the MAE measures the average size of the absolute size of the differences between the predicted values and the actual observations, visualizing the average level of prediction error. The

**Table 3 Assessment indicator system.**

| Indicators | Definition | Formula | |
|---|---|---|---|
| MSE | Mean square error | $MSE = \dfrac{1}{n}\sum_{i=1}^{n}\left(y_i - \widehat{y_i}\right)^2$ | (24) |
| MAE | Mean absolute error | $MAE = \dfrac{1}{n}\sum_{i=1}^{n}\left|y_i - \widehat{y_i}\right|$ | (25) |
| MAPE | Mean absolute percentage error | $MAPE = \dfrac{1}{n}\sum_{i=1}^{n}\left|\dfrac{y_i - \widehat{y_i}}{y_i}\right|*100\%$ | (26) |
| $R^2$ | Mean absolute percentage error | $R^2 = \dfrac{\sum_{i=1}^{n}w_i\left(\widehat{y_i} - \bar{y}_i\right)^2}{\sum_{i=1}^{n}w_i\left(y_i - \bar{y}_i\right)^2}$ | (27) |

MAE visualizes the average level of prediction error by measuring the average size of the absolute difference between the predicted value and the actual observed value. On the other hand, the $R^2$ metric quantifies the goodness of fit or correlation between the model's predictions and observations. The formula for each indicator is shown in Table 3.

# EXPERIMENTAL RESULTS AND ANALYSIS

## Model parameter setting

To verify the prediction accuracy of the COA-XGBoost-CEEMDAN model, various comparison algorithms are used to analyze the prediction effect. For the base model prediction, GBDT, support vector machine (SVR), and XGBoost models are selected to compare and analyze the prediction effect of COA-XGBoost; for the residual correction combination model prediction, the residual sequences generated by GWO-XGBoost are compared and decomposed using VMD and CEEMDAN methods. The parameter settings and optimal parameters of each model are shown in Table 4, and the rest of the parameters are set using Python default parameters.

## Experiment 1: Results of carbon price prediction in Hubei Province

The evaluation metrics of the established prediction framework and other prediction models are shown in Table 5 and Fig. 6, and the comparison of the predicted values of different models is shown in Fig. 7. The values with the best results in Table 5 will be bolded, and Fig. 6 visualizes the prediction performance of each model in the form of bar charts. The experimental data show that the model proposed in this article outperforms the other compared models in all cases. The following conclusions can be drawn.

(1) The prediction model proposed in this article outperforms other models in any comparison of evaluation metrics. According to the results listed in Table 5, the MSE, MAE, MAPE, and $R^2$ of the designed forecasting framework are 2.0573e−06, 0.000268, 0.00003, and 0.9999, respectively.

(2) When comparing the single model XGBoost and XGBoost with the cheetah optimization algorithm added, it is found that the model with the Cheetah optimization algorithm added is more accurate. The four evaluation metrics of MSE, MAE, MAPE, and

**Table 4 Model parameter setting.**

| Models | Parameterization | Hubei optimized parameters | Guangzhou optimized parameters |
|---|---|---|---|
| GBDT | (n_estimators, learning_rate, max_depth, subsample) | (1,000, 0.1, 6, 0.6) | (1,000, 0.1, 6, 0.8) |
| SVR | (Kernel, C, epsilon) | ('rbf', 1.0, 0.01) | ('rbf', 1.0, 0.01) |
| COA | lb/ub= learning_rate, subsample, colsample_bytree, max_depth, Alpha) | lb = [0.01, 0.5, 0.5, 3, 0] ub = [0.3, 1, 1, 10, 1] | lb = [0.01, 0.5, 0.5, 3, 0] ub = [0.3, 1, 1, 10, 1] |
| XGBoost | (learning_rate, subsample, colsample_bytree, max_depth, Alpha) | (0.1, 0.8, 0.8, 5, 0) | (0.1, 0.8, 0.4, 5, 0) |
| COA-XGBoost | (learning_rate, subsample, colsample_bytree, max_depth, Alpha) | (0.29996261, 0.9667642, 0.89935618, 9.12183629, 0.01051329) | (0.2929408, 0.98066281, 0.94465118, 9.50070627, 0.02067619) |

**Table 5 The forecasting results for theHubei dataset.**

| Models | MSE | MAE | MAPE | $R^2$ |
|---|---|---|---|---|
| GBDT | 0.3337 | 0.3846 | 1.0519 | 0.9953 |
| SVR | 0.2492 | 0.3680 | 1.0050 | 0.9965 |
| XGBoost | 0.1536 | 0.2788 | 0.7580 | 0.9979 |
| WOA-XGBoost | 0.0882 | 0.2150 | 0.5862 | 0.9987 |
| COA-XGBoost | 2.4996e-06 | 0.0011 | 0.0029 | 0.9999 |
| COA-XGBoost-VMD | 2.4879e-06 | 0.0011 | 0.00003 | 0.9999 |
| **COA-XGBoost-CEEMDAN** | **2.0573e−06** | **0.000268** | **0.00003** | **0.9999** |

**Note:**
The best result is shown in bold.

$R^2$ for the XGBoost model are 0.1536, 0.2788, 0.7580 and 0.9979, respectively. The values of COA-The values of the four evaluation metrics of XGBoost are 2.4796e−06, 0.0011, 0.0029, and 0.9999, which are 99.9%, 99.6%, 99.6%, and 0.2% respectively, compared to the metrics of XGBoost. It can be seen that the accuracy of the model with the addition of the optimization algorithm has been improved to some extent.

(3) The single model without the decomposition algorithm added is compared with the model incorporating the decomposition algorithm, which has better prediction results. The COA-XGBoost model with the CEEMDAN decomposition algorithm incorporated has values of 2.0573e−06, 0.000268, 0.00003, 0.9999, and the percentage improvement of the four metrics are 17.0%, 75.6%, 99.0%, and 0.000001%, which is a substantial improvement in accuracy. The contribution of the CEEMDAN algorithm to the prediction results is shown.

## Experiment 2: Results of carbon price prediction in guangzhou

To prove the robustness of the model, a further carbon market in Guangzhou will be selected for an experimental study. Taking the relevant data of the Guangzhou carbon market as a sample, the prediction results of each model are shown in Table 6 and Fig. 8, and the line graph comparing the predicted values of different models is shown in Fig. 9. It can be summarized from numeral charts that Experiment 2 has the same conclusion as

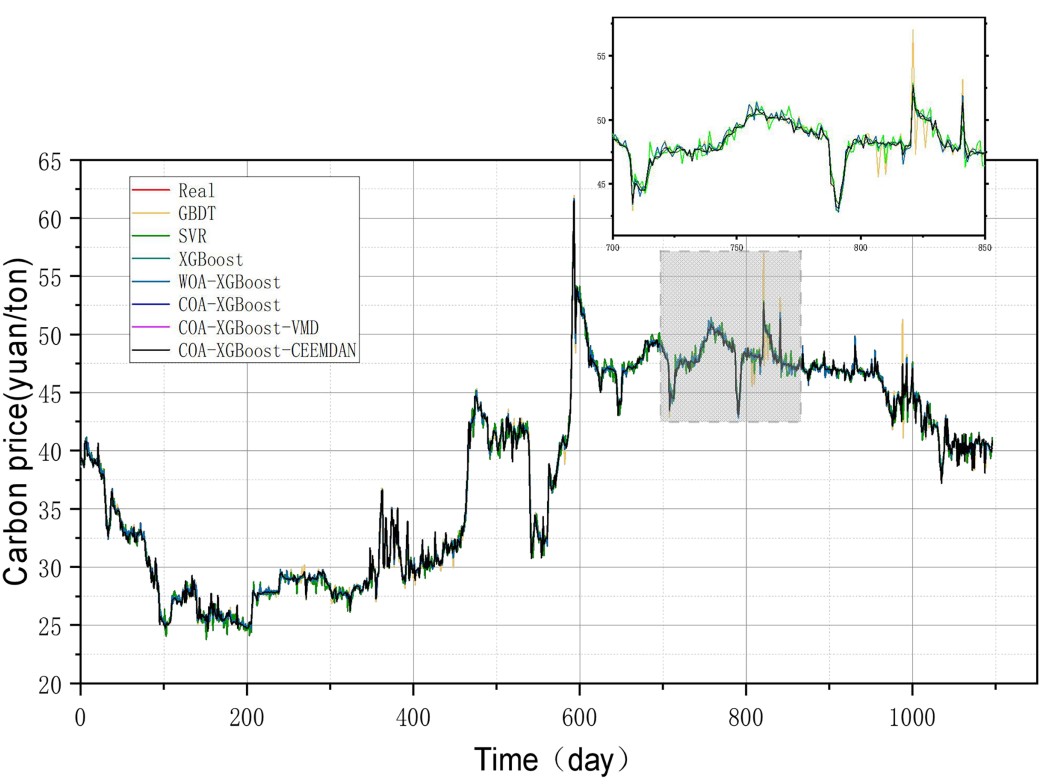

**Figure 6 Performance of different models on the Hubei test set predictions.**

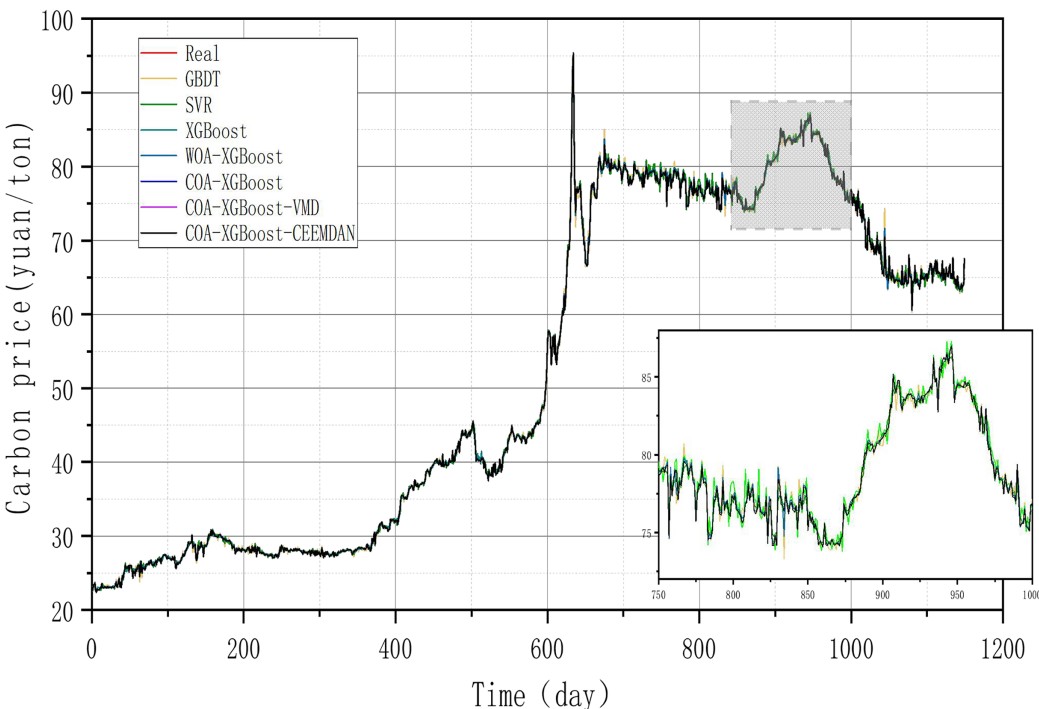

**Figure 7 Performance of different models on the Guangzhou test set predictions.**

**Table 6 The forecasting results for theGuangzhou dataset.**

| Models | MSE | MAE | MAPE | $R^2$ |
|---|---|---|---|---|
| GBDT | 0.2231 | 0.3200 | 0.6436 | 0.9996 |
| SVR | 0.1960 | 0.3073 | 0.5964 | 0.9996 |
| XGBoost | 0.1259 | 0.2666 | 0.5743 | 0.9996 |
| WOA-XGBoost | 0.0371 | 0.1147 | 0.2238 | 0.9999 |
| COA-XGBoost | 3.6450e−06 | 0.0013 | 0.0031 | 0.9999 |
| COA-XGBoost-VMD | 3.6351e−06 | 0.0013 | 0.00003 | 0.9999 |
| **COA-XGBoost-CEEMDAN** | **1.4476e−10** | **0.000009** | **0.0000002** | **0.9999** |

**Note:**
The best result is shown in bold.

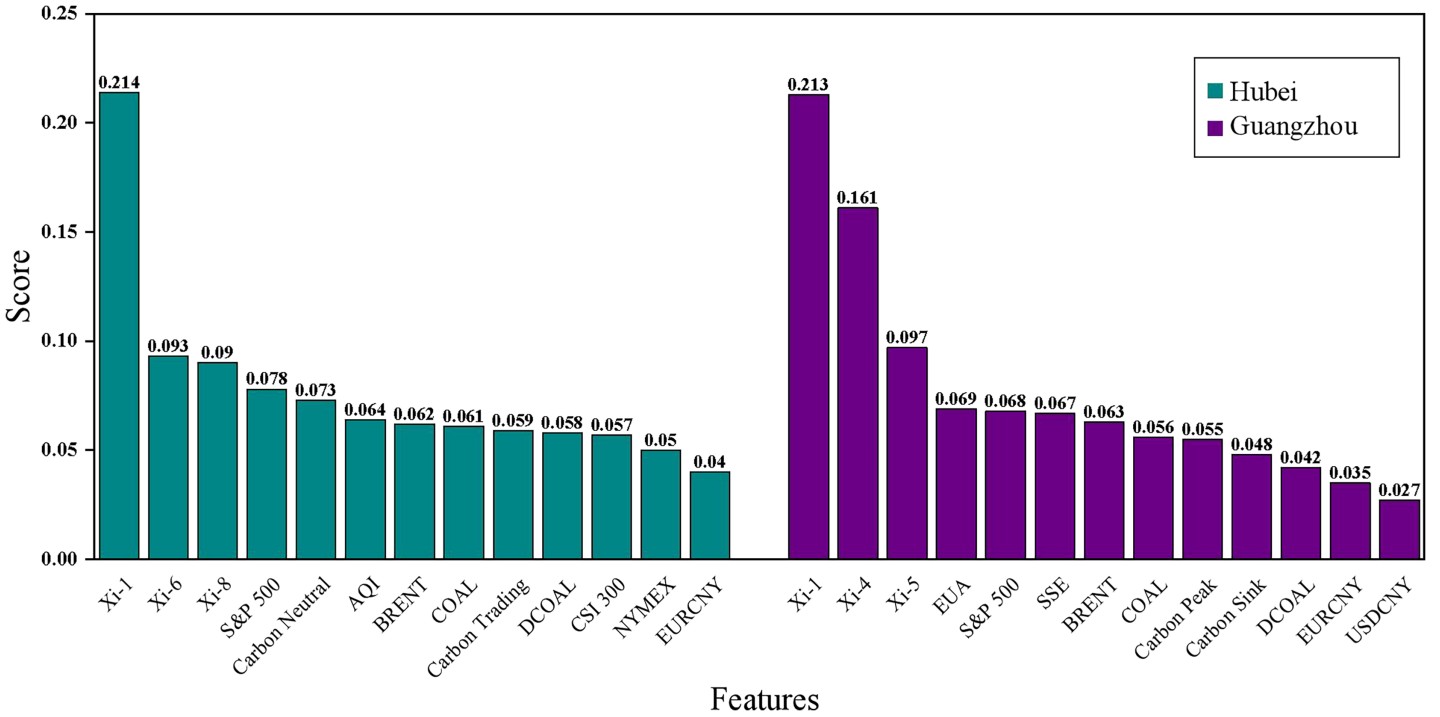

**Figure 8 Feature importance histogram.**     

Experiment 1. It shows that the model proposed in this article combines the advantages of optimization algorithm, decomposition algorithm, and feature selection, and performs well in all aspects.

## Importance analysis of carbon price characteristics

Characteristic importance analysis can help the government to make scientific decisions in various aspects such as an in-depth understanding of the energy market, optimizing policy design, and formulating cross-cutting policies by identifying the factors that have an important impact on the target value of the forecast (*Chang & Park, 2023*). In this article,

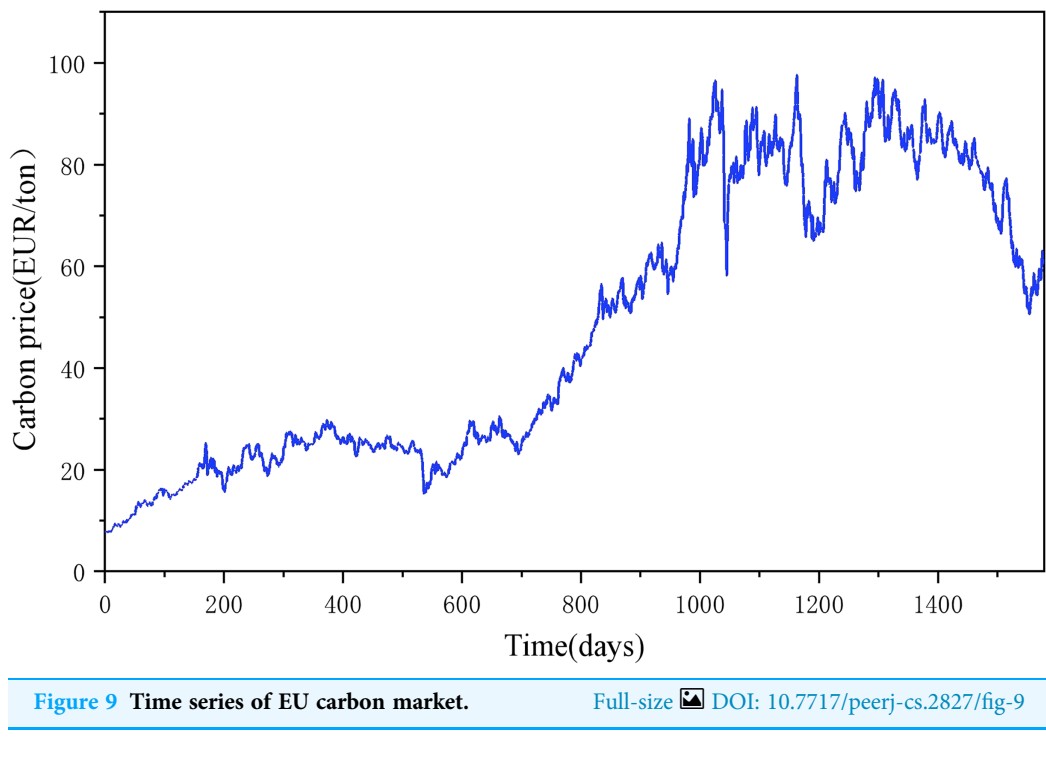

**Figure 9 Time series of EU carbon market.**

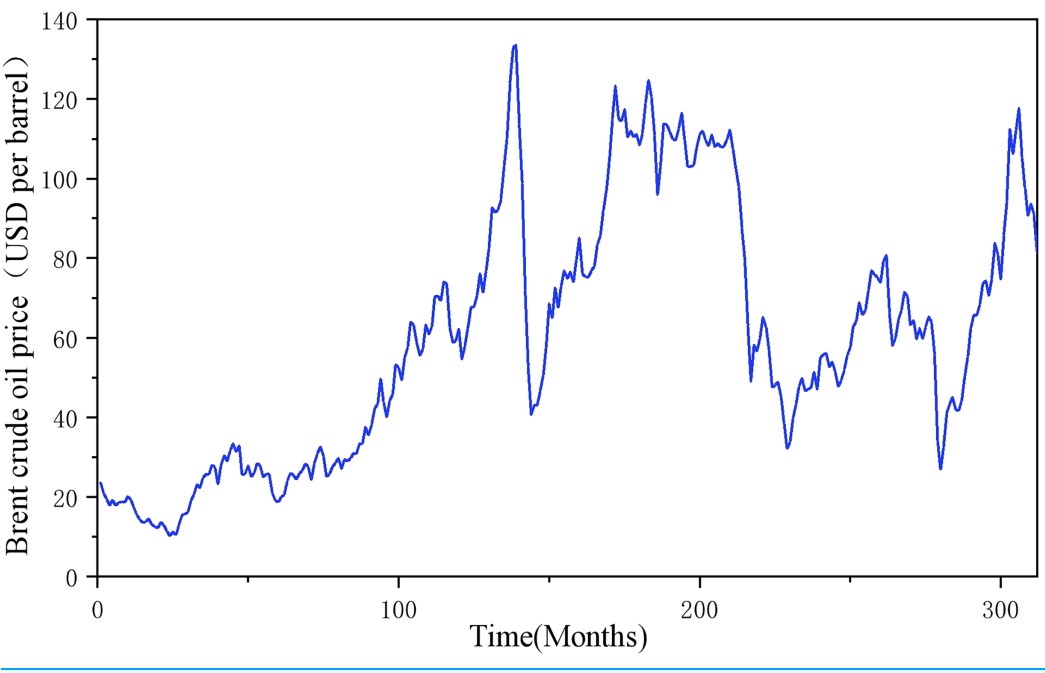

**Figure 10 Time series of Brent crude oil prices.**

the Lasso regression model is used for feature importance analysis. The results of the feature importance ranking of each model are shown in Fig. 10 and Table 7. Analyzing the feature ranking in Table 7, it is found that the historical carbon price is the best data source for predicting the carbon price in these two carbon markets, which is because the price time series is a comprehensive external manifestation of the intrinsic complexity of the

**Table 7 Ranking of importance of carbon price features.**

| Hubei | | Guangzhou | |
|---|---|---|---|
| **Feature** | **Score** | **Feature** | **Score** |
| $x_{i-1}$ | 0.214 | $x_{i-1}$ | 0.213 |
| $x_{i-6}$ | 0.093 | $x_{i-4}$ | 0.161 |
| $x_{i-8}$ | 0.09 | $x_{i-5}$ | 0.097 |
| S&P 500 | 0.078 | EUA | 0.069 |
| Baidu Search Index-Carbon Neutral | 0.073 | S&P 500 | 0.068 |
| AQI | 0.064 | SSE | 0.067 |
| BRENT | 0.062 | BRENT | 0.063 |
| COAL | 0.061 | COAL | 0.056 |
| Baidu Search Index-Carbon Trading | 0.059 | Baidu Search Index-Carbon Peak | 0.055 |
| DCOAL | 0.058 | Baidu Search Index-Carbon Sink | 0.048 |
| CSI 300 | 0.057 | DCOAL | 0.042 |
| NYMEX | 0.05 | EURCNY | 0.035 |
| EURCNY | 0.04 | USDCNY | 0.027 |

market, which contains important information about the past market performance and price fluctuations, and can help researchers to analyze and predict through these historical price data. Additionally, it can be observed that the settlement price of Brent crude oil futures in the energy factors and the S&P 500 index in the macroeconomic factors rank among the top in both datasets, making them key external factors in carbon price prediction.

## DISCUSSION

### Experiments on the application of the model
#### EU carbon price forecasts

To explore the generalization ability of the proposed model and the generalizability of the study, price forecasts for the EU carbon market are added. An indicator system is established by considering international carbon prices, macroeconomics, energy prices, climatic conditions, macro-policy and emerging media indices, and historical prices. The daily trading data of EUA from January 1, 2018 to March 31, 2024, is selected for forecasting. There are 1,577 data in total. As shown in Fig. 11, the carbon price experienced a typical price change pattern during the training period, including the three stages of rise, fall, and oscillation. The results of the metric calculations for each of the models are shown in Table 8, with the best data in the table shown in bold.

According to Table 8, the following conclusions can be obtained. First, COA-XGBoost with optimization algorithm added has an excellent performance in a single machine learning model, and the evaluation indexes are improved by 99.99%, 99.39%, 99.34%, and 0.05% respectively compared with XGBoost without optimization algorithm added. From this, it can be concluded that using COA to optimize the parameters of XGBoost can improve the prediction accuracy. Secondly, the accuracy of the hybrid algorithm with the

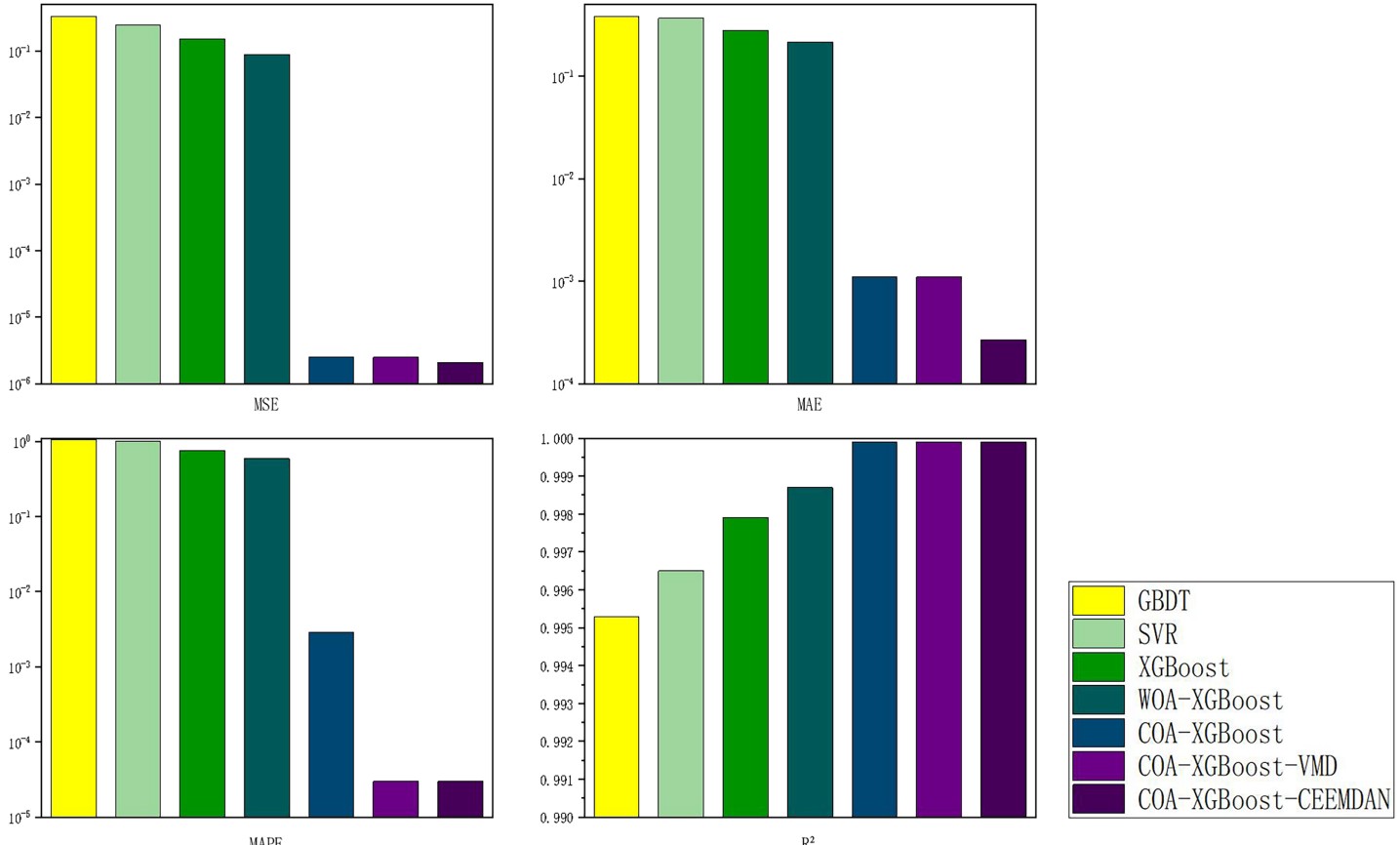

**Figure 11  Map of carbon market evaluation indicator results in Hubei Province.**

**Table 8  The forecasting results for the EU dataset.**

| Models | MSE | MAE | MAPE | R² |
|---|---|---|---|---|
| GBDT | 0.8679 | 0.5725 | 1.0009 | 0.9986 |
| SVR | 0.5923 | 0.5087 | 0.9065 | 0.9991 |
| XGBoost | 0.3620 | 0.4593 | 0.9585 | 0.9994 |
| WOA-XGBoost | 0.2370 | 0.3284 | 0.5833 | 0.9996 |
| COA-XGBoost | 1.6563e−05 | 0.0028 | 0.0063 | 0.9999 |
| COA-XGBoost-VMD | 1.6417e−05 | 0.0028 | 0.00006 | 0.9999 |
| **COA-XGBoost-CEEMDAN** | **1.2653e−07** | **0.0003** | **0.000006** | **0.9999** |

Note:
The best result is shown in bold.

addition of VMD and CEEMDAN is significantly improved, CEEMDAN presents a small increase compared to the VMD basis, and the evaluation indexes of COA-XGBoost-CEEMDAN reach 1.2653e−07, 0.0003, 0.000006, and 0.9999, respectively. Through the preliminary decomposition of the predicted residual sequences, effective information is extracted, which further improves the accuracy of prediction.

In summary, the model proposed in this article shows the same excellent results in the prediction of the EU carbon trading market compared with several other models, which proves the generalization ability of this model and can be better applied to new data, indicating its universality and reliability.

### Oil price forecasts

To further verify the generalization performance of this model in heterogeneous energy markets, this study extends the prediction scenario to the international crude oil market. Based on the multi-scale coupling characteristics of the global energy market, a system of indicators covering commodity attributes, economic factors, alternative energy sources and geopolitical factors, as well as its own historical prices, is constructed. All indicators are based on monthly data with the time interval from January 1997 to December 2022, totaling 212 data, as shown in Fig. 12. The Lasso-COA-XGBoost-CEEMDAN model proposed in this article is used to predict the forecast of Brent oil price, and the model performance is quantified by a quadratic evaluation system: MSE = 0.1553, MAE = 0.3544, MAPE = 0.0058 and $R^2$ = 0.9997. The results show that the model also exhibits excellent predictive performance in crude oil price forecasting.

## Comparison between the proposed model and other models

In recent years, there have been numerous studies on carbon prices. To prove the validity of the proposed model, this article compares other hybrid models for carbon price forecasting. The comparison models are described as follows:

*Ke et al. (2023)* proposed multi-decomposition-XGBoost model. It combines the results of the first and second decompositions based on sample entropy, then performs another round of decomposition and uses the XGBoost prediction model to make predictions, and finally, summarizes the results to obtain the carbon price combination prediction.

*Hu & Cheng (2023)* developed the SD-RE-MIC-SSA-HKELM-Ensemble model to forecast carbon price. It utilizes variational mode decomposition (VMD) to decompose the carbon price into several modes and then reconstructs these modes using polar entropy. The multifactor HKELM optimized by the sparrow search algorithm is used to forecast the reconstructed subsequences, and the main external factors and the historical time series data of carbon price innovatively selected by the information coefficient maximization method are used as the input variables of the forecasting model. Finally, a nonlinear integrated learning method is introduced to determine the residual term and the predicted value of the final carbon price.

The above hybrid models present innovative approaches in the field of carbon price forecasting research and advance the field of carbon price forecasting. By comparing with these models, the performance of the proposed model in predicting carbon price can be evaluated. The proposed model and the compared models are compared by three evaluation indexes, RMSE, MAE, and MAPE, and the results are shown in Table 9, with the optimal results shown in bold. Meanwhile, to ensure the rigor of the comparison, the time horizon of the prediction is adjusted to be consistent. The time range of the Hubei

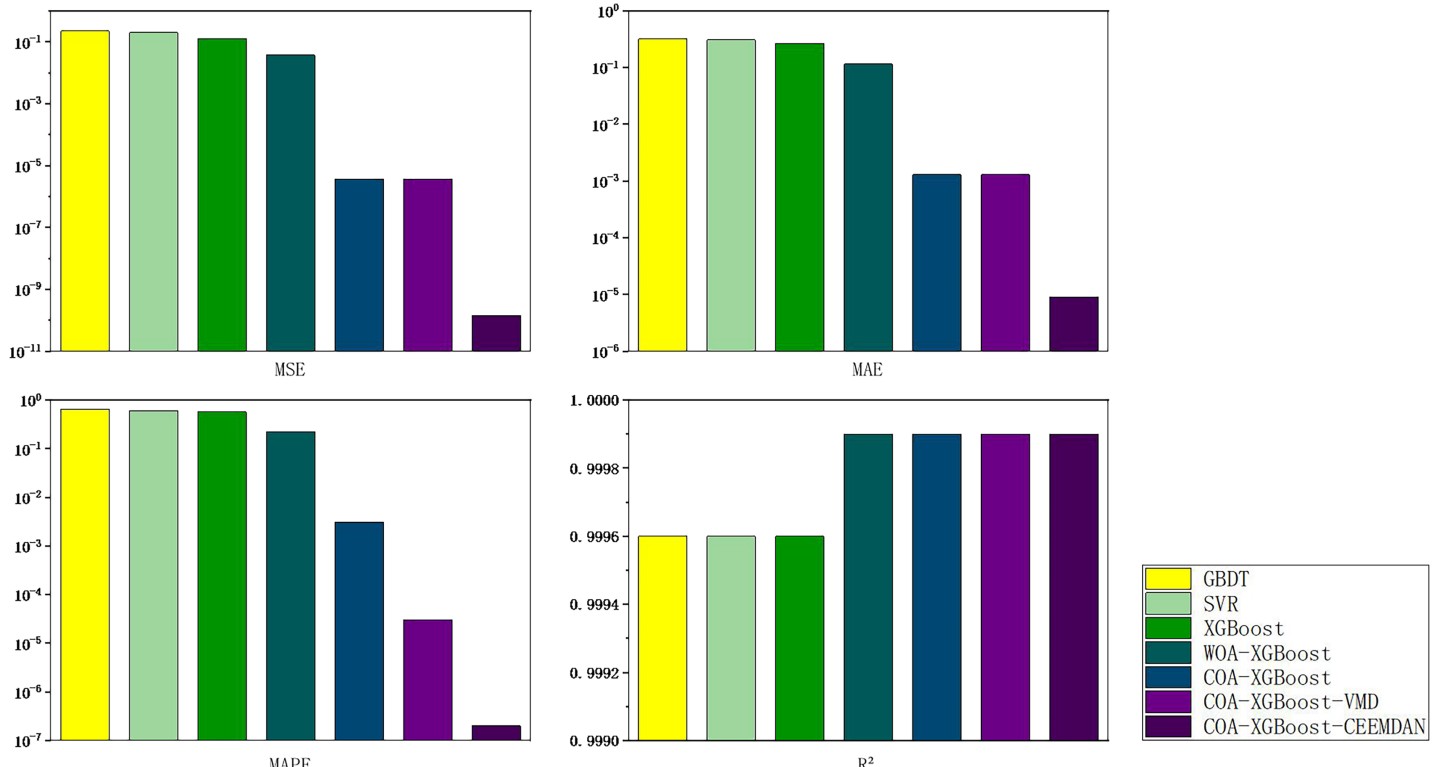

**Figure 12 Hubei carbon market evaluation indicator results chart.**     

**Table 9 Comparison of prediction results between this model and other mixed models.**

| Carbon market | Model | RMSE | MAE | MAPE |
|---|---|---|---|---|
| Hubei | Multi-decomposition-XGBoost | 0.748 | 0.505 | 1.745% |
| | **Lasso-COA-XGBoost-CEEMDAN** | **0.002** | **0.0004** | **0.16%** |
| Guangzhou | SD-RE-MIC-SSA-HKELM-Ensemble | 0.1716 | 0.1218 | 0.26% |
| | **Lasso-COA-XGBoost-CEEMDAN** | **0.004** | **0.0027** | **0.18%** |

Note:
The best results are shown in bold.

carbon market is from August 1, 2018 to November 13, 2020, and the time range of the dataset of the Guangzhou carbon market is from January 3, 2017 to February 28, 2022.

After comparative analysis with other models, it is found that the model proposed in this study presents superior performance performance. As can be seen from Table 9, hybrid models have been widely used and highly matured in the field of carbon price forecasting, by combining different types of forecasting models to better integrate the advantages of the models and improve the forecasting accuracy. In the comparison of forecasting in the Hubei carbon market, the RMSE, MAE, and MAPE of the model proposed in this article reach 0.002%, 0.0004%, and 0.16%, respectively, and the values of all evaluation indexes are better than those of the comparison model. In the comparison experiment of the Guangzhou carbon market, the comparative model's indicators are 0.1716%, 0.1218%, and 0.26%, and the RMSE and MAE performance is not as good as the

performance of the model proposed in this article, though. None of the above comparative models have made split-frequency predictions of the residual series or considered the influence of extreme events, which are possible reasons why the prediction accuracy cannot be improved. None of the above comparative models performs split-frequency prediction on the residual series, ignoring the rich information contained in the residuals, and does not take into account the impact of extreme events, which makes the model lack of adaptability and robustness in the face of such special cases, and these are the reasons that may lead to the failure to improve the prediction accuracy.

## Limitations and future directions of work

In this article, a novel carbon price prediction method is proposed, which significantly improves the prediction accuracy and thus assists governments, enterprises, and investors in making more accurate decisions. Although the results of the study show good prediction results, there are still some limitations. In terms of the model: (1) The heuristic optimization algorithm used is more complicated in terms of parameter configuration. Determining the appropriate parameters not only requires a lot of time and computational resources, but also this complex parameter adjustment process restricts the enhancement of the model's intelligence level to a certain extent, which reduces the convenience and efficiency of the model application. (2) Although the CEEMDAN method is used to decompose the residual series, it is limited to a single decomposition, which may not be able to fully capture the deeper nonlinear features. In the study of the carbon price problem: (1) This study only realizes a single-step forecast and fails to comprehensively analyze the future carbon price trend from a more macroscopic perspective. (2) The characteristics of carbon price influencing factors over time are neglected.

Based on the above limitations, future research work can be carried out in the following directions: firstly, in-depth research should be carried out on the optimization algorithm and model streamlining, a more intelligent and convenient prediction model should be constructed, the automation of the parameter setting of the heuristic optimization algorithm should be realized, and the complexity of the experimental steps should be reduced, so as to improve the operation efficiency and application popularization of the model. Secondly, the introduction quadratic decomposition in the residual sequence processing should be attempted to fully reveal the deep nonlinear relationship and make the model prediction more accurate. Finally, the carbon price prediction method in this study can be extended and applied to other related fields, such as energy market prediction and climate change risk management, which will verify the generalizability and application value of the research results.

## Policy recommendations

As one of the world's largest carbon emitters, establishing a robust carbon market in China will have a positive impact on reducing global carbon emissions. The following related policy recommendations are proposed based on the research findings of this article:

(1) As a key indicator for predicting future carbon prices, historical carbon prices provide extremely valuable information about their trends and patterns. Therefore, the

government should first set up a carbon market reserve mechanism to put in or recover carbon quotas at the right time according to the market condition, to ensure the stability of carbon market prices and prevent drastic fluctuations. In addition, the government should also build a complete carbon market database, including historical prices, trading volume, and emission data of various carbon markets, to facilitate researchers' access to relevant information for in-depth analysis and research.

(2) Fluctuations in energy prices are directly related to the production costs and energy choices of enterprises, which have a complex impact on carbon prices. For the Hubei carbon market, coal prices and power coal prices have a high impact score on carbon price forecasts. For the Guangzhou carbon market, crude oil prices and power coal prices have higher impact scores on carbon price forecasts. The government should take several measures to reduce the burden of production costs on enterprises when energy prices rise. Firstly, the government should support companies in improving energy efficiency and technology upgrades through financial subsidies and soft loans. Second, it should actively promote the development of clean energy, increase the supply of clean energy, and reduce reliance on coal and oil, thereby reducing the volatility of carbon prices and their impact on traditional energy prices. When energy prices fall, the government needs to review and adjust the carbon quota allocation mechanism to ensure that the carbon market still incentivizes companies to adopt low-carbon production and business strategies. For the Guangzhou carbon market in particular, to ensure the accuracy of carbon price forecasts and promote sustainable economic development, the government needs to pay close attention to the price movements of natural gas and coal.

(3) On the economic front, the Hubei carbon market should monitor the price movements of the S&P 500, the EUR-RMB exchange rate, and the CSI 300, and the Guangzhou carbon market should pay attention to the price movements of the USD-RMB exchange rate, the S&P 500, the SSE Composite Index and the EUR-RMB exchange rate, which are of reference value in accurately forecasting carbon market prices. When these indices rise, it indicates optimism in the stock market and expansion of economic activity, which may increase demand for the carbon market. The government should capitalize on periods of economic prosperity to strengthen training and education efforts related to the carbon market, raise awareness of the carbon market among enterprises and investors, and promote active trading in the market. Additionally, when indicators are falling, the government should take measures to stimulate economic growth and promote the sustainable development of a low-carbon economy. To lessen the detrimental effects of exchange rate swings on the economy and the energy market, the government must also keep the currency rate reasonably constant.

(4) To better grasp the public's concern and the trend of public opinion on the carbon market and related issues, the government should make full use of big data tools such as the Baidu index. The Hubei carbon market should monitor the trends of the keywords "carbon neutral" and "carbon trading", and the Guangzhou carbon market should pay attention to the trends of the keywords "carbon peak" and "carbon sink". The Guangzhou carbon market should focus on the trends of the keywords "carbon peak" and "carbon sink", which have reference values for accurate prediction of carbon market price. By

monitoring the trend of keyword searches related to the carbon market, the government can keep abreast of changes in the public's interest in carbon emissions and carbon trading, and provide a basis for developing more targeted publicity and education programs. In addition, changes in the Baidu index can also reflect market expectations and public sentiment, helping to warn of potential market fluctuations in advance and assisting policymakers in making more scientific and rational decisions.

## CONCLUSION

Maintaining the robust growth of China's carbon trading market requires the development of a high-precision prediction model that can reliably forecast the price of carbon trading in China. In this study, the CEEMDAN decomposition method is introduced into the problem of correction of residual series, and further individual prediction of the eigenmode function generated by CEEMDAN is performed, and finally, the COA-XGBoost-CEEMDAN integrated learning combination prediction model is established. In this study, seven prediction models are utilized to study the carbon price prediction problem in the Hubei and Guangzhou carbon markets, and four evaluation indexes are introduced to discuss the prediction accuracy and stability of each model in different datasets. The empirical analysis leads to the following conclusions:

(1) Out of all the benchmark models, the forecasting framework suggested in this research performs the best. Taking the carbon price prediction in Hubei as an example, the MSE, MAE, MAPE, and $R^2$ of the model are 2.0573e−06, 0.000268, 0.00003, and 0.9999, respectively. And verified in other experiments, which demonstrates the superiority and universality of the model in carbon price prediction.

(2) The models applicable to different frequencies are used for the next prediction of the decomposed carbon price residuals. Firstly the introduction of the decomposition algorithm can greatly reduce the volatility and randomness of the data, thus improving the performance of the prediction model. Meanwhile, it can be concluded through experiments that the decomposition effect of CEEMDAN is better than that of VMD. Secondly, the residual terms are reconstructed into high frequency and low frequency, and the COA-XGBoost model and SVR model applicable to each frequency are selected to predict the reconstructed high frequency and low frequency for the sharp fluctuation of the high-frequency sequence and the smaller fluctuation and randomness of the low-frequency sequence, respectively.

(3) An integrated prediction method for dynamic carbon price decomposition based on a rolling time window should be constructed, avoiding the data leakage problem while ensuring the timeliness and effectiveness of the prediction model.

(4) Considering the index system of the Baidu index makes a significant contribution to the research of carbon price prediction. Carbon price is complex, and there are many factors affecting its price fluctuation and a wide range of factors. Considering only the historical carbon price data is not enough to accurately predict the carbon price; therefore, this article constructs a hybrid prediction framework that integrates multiple influencing factors of feature selection and preprocesses all input data to improve the prediction performance.

The model proposed in this article improves the prediction accuracy and has significant value and prospect to provide strong support for the stable development of the carbon market, sustainable economic growth and the realization of global climate goals. However, there are still limitations in the study. Firstly, in the study of the carbon price problem, only a single-step prediction is realized, and the future trend is not comprehensively analyzed; also, the time-varying characteristics of the factors affecting the carbon price are ignored, and its complexity and numerous parameter options lead to the fact that the model requires a longer period of time to run, even though it performs better in the prediction. Future work needs to strengthen the research on model streamlining, reduce the complexity of experimental steps, improve the model prediction accuracy, and also extend the prediction model to other fields to further prove the value of the research results.

### Funding
This research was supported by the Natural Science Foundation of Henan Province (Grant no. 242300421257). The funders had no role in study design, data collection and analysis, decision to publish, or preparation of the manuscript.

### Grant Disclosures
The following grant information was disclosed by the authors:
Natural Science Foundation of Henan Province: 242300421257.

### Competing Interests
The authors declare that they have no competing interests.

### Author Contributions
- Yonghui Duan conceived and designed the experiments, authored or reviewed drafts of the article, and approved the final draft.
- Yingying Fan conceived and designed the experiments, performed the experiments, analyzed the data, performed the computation work, prepared figures and/or tables, and approved the final draft.
- Xiang Wang performed the experiments, authored or reviewed drafts of the article, and approved the final draft.
- Kaige Liu analyzed the data, authored or reviewed drafts of the article, supervision, and writing-review, and approved the final draft.
- Xiaotong Zhang performed the computation work, authored or reviewed drafts of the article, supervision, and writing-review, and approved the final draft.

### Data Availability
The Hubei dataset, Guangzhou dataset, and EU dataset from CHOICE Financial Terminal is available at figshare: Fan, Yingying (2025). Daily Carbon Price Data from

Hubei, Guangzhou, and EU Emissions Trading Systems (2018–2024). figshare. Dataset. https://doi.org/10.6084/m9.figshare.28682684.v1.

## Supplemental Information

Supplemental information for this article can be found online at http://dx.doi.org/10.7717/peerj-cs.2827#supplemental-information.

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
