# Peer review of "Dynamic prediction of carbon prices based on the multi-frequency combined model"

_PeerJ Computer Science, doi:10.7717/peerj-cs.2827_

## Round 0.1 · original submission · Major Revisions

The paper addresses a relevant topic with a promising impact on the sustainability of chinese (and global) industry. However, the manuscript lacks a robust discussion of the proposed method w.r.t. state of the art methods, as well as clarity in motivating the study (Introduction) and quality of the presentation (e.g., image quality).

Therefore, the current version of the manuscript does not show the proper rigorous presentation and contents that are required for publication. The authors are encouraged to carefully read the reviewers' comments and to revise their manuscript, accordingly. It may be required to add experiments and new results to support COA algorithm's effectiveness.

Reviewer 1 ·

Basic reporting

no comment

Experimental design

no comment

Validity of the findings

no comment

Additional comments

Authors used alot acroymn first without full meaning ensure that meaning are used before continuing with acryomn e.g COA, XGBoost e.t.c

Authors should review introduction section to include more recent research of Machine learning ehanced with optimization algorithm such as


The image quality require enhancement at least 300dpi

Avoid personal pronouns e.g we or our maintain formal and academic tone. Ensure formal and academic tone in the entire manuscript

Include quantitative description of findings in the abstract

Phrases such as e,g "As a key player in the global response to climate change", "has become a hot topic of research" do not demonstrate
shcolary articulation correct such sentences

The motivation and contribution is unclear in the introduction

There's no experiment to test COA algorthim before applying to the problem

Discuss the limitation of the study


#


1. The manuscript includes multiple instances where acronyms such as COA and XGBoost are used without first providing their full meanings. The authors must ensure that all acronyms are clearly defined upon first usage before continuing with their abbreviated forms.

2. The introduction should be revised to incorporate more recent research on machine learning enhanced with optimization algorithms.

3. The quality of the images must be improved to a resolution of at least 300 dpi to ensure clarity and compliance with publication standards.

4. The authors should avoid the use of personal pronouns such as "we" or "our." The entire manuscript should maintain a formal and academic tone.

5. The abstract should include a quantitative description of the findings to provide a clearer and more precise overview of the study's outcomes.

6. Phrases such as "As a key player in the global response to climate change" or "has become a hot topic of research" lack scholarly articulation. These sentences should be revised to reflect a more academic and professional style.

7. The motivation and contributions of the study are unclear in the introduction. This section should be refined to clearly articulate the novelty and significance of the work.

8. There is no evidence of experiments conducted to validate the COA algorithm before applying it to the problem. The authors should include a dedicated experiment or discussion to justify its application.

9. The manuscript does not address the limitations of the study. A section discussing potential constraints, challenges, or areas for future improvement should be included.

Cite this review as

·

Basic reporting

The topic is interesting and with possible applicability. However, I have some comments and suggestions which could greatly improve the manuscripts quality and these are:
1- The authors should highlight why COA is selected despite here are several other metaheuristic methods were available
2- Environmental setting used to implement the COA, and other machine learning should be added
3- Some new works using and applicable studies should be added.
4- What is the advantage over established techniques.
5- Conclusion is very weak. It should summarize the research, mention the best scores achieved, mention limitations and finally provide clear directions for the future research in this domain

Experimental design

Environmental setting used to implement the COA, and other machine learning should be added

Validity of the findings

The results are good

Additional comments

N/A

---

## Round 0.2 · accepted · Accept

The authors have fulfilled all reviewers' comments and now the manuscript is ready for publication.

Reviewer 1 ·

Basic reporting

no comment

Experimental design

no comment

Validity of the findings

no comment

Additional comments

no comment

Cite this review as

·

Basic reporting

The authors addressed all the required comments

Experimental design

good

Validity of the findings

good

Additional comments

no comments